# The Key Role of Active Sites in the Development of Selective Metal Oxide Sensor Materials

**DOI:** 10.3390/s21072554

**Published:** 2021-04-06

**Authors:** Artem Marikutsa, Marina Rumyantseva, Elizaveta A. Konstantinova, Alexander Gaskov

**Affiliations:** 1Chemistry Department, Moscow State University, 119991 Moscow, Russia; roum@inorg.chem.msu.ru (M.R.); gaskov@inorg.chem.msu.ru (A.G.); 2Physics Department, Moscow State University, 119991 Moscow, Russia; liza35@mail.ru; 3Faculty of Nano-, Bio-, Information and Cognitive Technologies, Moscow Institute of Physics and Technology, 141700 Dolgoprudny, Russia; 4National Research Center “Kurchatov Institute”, 123182 Moscow, Russia

**Keywords:** active sites, metal oxide semiconductor, gas sensor, gas-solid interaction, selectivity, nanoparticles

## Abstract

Development of sensor materials based on metal oxide semiconductors (MOS) for selective gas sensors is challenging for the tasks of air quality monitoring, early fire detection, gas leaks search, breath analysis, etc. An extensive range of sensor materials has been elaborated, but no consistent guidelines can be found for choosing a material composition targeting the selective detection of specific gases. Fundamental relations between material composition and sensing behavior have not been unambiguously established. In the present review, we summarize our recent works on the research of active sites and gas sensing behavior of *n*-type semiconductor metal oxides with different composition (simple oxides ZnO, In_2_O_3_, SnO_2_, WO_3_; mixed-metal oxides BaSnO_3_, Bi_2_WO_6_), and functionalized by catalytic noble metals (Ru, Pd, Au). The materials were variously characterized. The composition, metal-oxygen bonding, microstructure, active sites, sensing behavior, and interaction routes with gases (CO, NH_3_, SO_2_, VOC, NO_2_) were examined. The key role of active sites in determining the selectivity of sensor materials is substantiated. It was shown that the metal-oxygen bond energy of the MOS correlates with the surface acidity and the concentration of surface oxygen species and oxygen vacancies, which control the adsorption and redox conversion of analyte gas molecules. The effects of cations in mixed-metal oxides on the sensitivity and selectivity of BaSnO_3_ and Bi_2_WO_6_ to SO_2_ and VOCs, respectively, are rationalized. The determining role of catalytic noble metals in oxidation of reducing analyte gases and the impact of acid sites of MOS to gas adsorption are demonstrated.

## 1. Introduction

### 1.1. Issues in the Design of Metal Oxide Sensors

Nanomaterials based on metal oxide semiconductors (MOS) have been extensively investigated for use in resistive gas sensors. The electronic properties of MOS are sensitive to chemical processes occurring at the surface, which enables the detection of gas molecules in the ppb-ppm concentration range in the environment [1,2,3]. Thermal stability in air, miniaturization, low cost and ability of integration into wireless devices are the main advantages of resistive sensors. The main drawback is the low selectivity, which is due to the sensing principle. Redox conversion of reducing or oxidizing gas molecules at the surface of MOS controls the thickness of a space charge region and electric conduction in the semiconductor [2,4]. Therefore, gas molecules with distinct compositions but similar redox properties are indistinguishable by the sensor response. A number of approaches to improve the selectivity of MOS sensors were elaborated. Those include technological ones, e.g., using preconcentration or filtering membranes [5,6], modulation of the operation temperature regime, use of multisensor arrays with appropriate data treatment algorithms (“electronic nose”) [7,8,9]. The fundamental approach is modification of the chemical composition of the sensor material. It can be performed by functionalization of MOS surfaces [9,10,11,12], bulk doping of MOS [13,14], use of *p*-type MOS and mixed-metal oxides instead of conventionally employed *n*-type MOS (SnO_2_, ZnO, WO_3_, TiO_2_, Ga_2_O_3_, In_2_O_3_, etc.) [15,16,17], or MOS-based composites with different additives, e.g., noble metals, metal oxides, graphene and carbon nanotubes, polymers [18,19,20,21]. Although there are numerous reviews available on MOS-based materials for gas sensors, a comprehensive materials design concept for the creation of sensors with predetermined selectivity for a desired analyte gas or a group of gases can hardly be found. In a recent review by Korotcenkov, et.al., the criteria for choosing the chemical composition of a gas-sensitive MOS were proposed [22]. For example, simple metal oxides conventionally used in sensors combine thermal stability (high melting and decomposition points above 1200 °C), and electronic configuration d^0^ for transition metal cations (in TiO_2_, WO_3_) or d^10^ for post-transition metal cations (in In_2_O_3_, ZnO, SnO_2_) that conditions the wide bandgap *n*-type semiconductor behavior. These cations may be partially reduced through the formation of oxygen vacancies, and the concentration of point defects in these metal oxides falls in the 10^16^–10^19^ cm^−3^ range, which is appropriate for the sensitivity of bulk conduction of localized charges at the surface [22]. However, no guidelines can be found for choosing the chemical composition of a sensor material. Even the choice between the listed simple MOS is an issue when the selectivity of analyte gas adsorption and redox interaction with the surface is considered. A more sophisticated problem is an appropriate selection of constituents when gas sensitive mixed-metal oxides or composites are elaborated. Besides the predictable effect on electronic properties of the semiconductor, the additives influence the adsorption capacity and chemical reactivity of the material surface, and the latter has often been disregarded in the gas sensor studies.

### 1.2. Gas Sensing Principles of Metal Oxide Sensors

The mechanism of gas sensing consists of three principle steps: (i) adsorption and redox interaction of gas molecules with the MOS surface (“reception”); (ii) transduction of the electric signal produced by the redox reaction from the surface to the bulk of MOS; and (iii) transduction of the electric signal through the network of contacting MOS particles [1,2,23]. The reception is a chemical process involving the active sites at MOS surface and is discussed in detail below.

#### 1.2.1. Oxygen Vacancy Formation and Oxygen Chemisorption

Oxygen vacancies are formed due to partial oxygen loss from the surface of oxides in which metals have the highest oxidation states for the respective elements group:O^2−^_(bulk)_ ↔ ½ O_2(g)_ + V_O_^2−^ ↔ ½ O_2(g)_ + V_O_^−^ + *e*^−^ ↔ ½ O_2(g)_ + V_O_ + 2 *e*^−^(1)

Electron donor states associated with oxygen vacancies (or, reduced metal cations) are available for oxygen chemisorption in air yielding chemisorbed species (Equation (2)), or restoring the lattice O^2−^ anions on the oxide surface (Equation (3)):O_2(g)_ + 4 *e*^−^ ↔ O_2_^−^_(ads)_ + 3 *e*^−^ ↔ 2 O^−^_(ads)_ + 2 *e*^−^ ↔ 2 O^2−^_(ads)_(2)
O_2(g)_ + 2 V_O_ + 4 *e*^−^ ↔ 2 O^2−^_(bulk)_(3)

As a result, in the subsurface region the donor states become less occupied by electrons, negative charge is localized at the chemisorbed oxygen and surface potential barrier arises leading to reduced conductivity of the *n*-type MOS decrease (Figure 1). The chemisorption equilibria described in Equation (2) shift to the right at increasing temperature, so that predominant chemisorbed oxygen species changes from molecular O_2_^−^ (160–200 °C) to atomic O^−^ (200–350 °C) and O^2−^ ones (above 350 °C). It was established for polycrystalline SnO_2_ using numerous spectroscopic techniques and indirect measurements [5,24,25], and the agreeing conclusions were done for other *n*-type MOS (ZnO, In_2_O_3_, WO_3_) using the conduction measurements under variable oxygen pressure [26,27,28].

There are two models describing the sensor response as the interaction of analyte gases with: (i) chemisorbed oxygen species (ionosorption model), and (ii) lattice oxygen anions at the oxide surface (oxygen vacancy model) [29,30].

#### 1.2.2. The Ionosorption Model of Sensor Response

Within the ionosorption model, sensor response to a reducing gas (Red) is due to oxidation of gas molecules by chemisorbed oxygen species (O_n_^m−^) [4,6,7,8,29,30]:Red_(g)_ + O_n_^m−^_(ads)_ → RO_n(g)_ + *m e*^−^(4)
where RO_n_ is the oxidation product. The removal of chemisorbed oxygen species releases the trapped electrons back to the semiconductor and decreases the surface potential barrier (Figure 2), which results in increasing conductivity. For example, the reducing gases H_2_, CO, H_2_S, NH_3_, and VOCs are commonly considered to be oxidized to H_2_O, CO_2_, H_2_O + SO_2_, H_2_O + N_2_, and H_2_O + CO_2_, respectively [2,4,5,31,32,33,34,35]. The surface reaction is described similarly as the mechanisms of heterogeneous oxidation catalysis within the Langmuir-Hinshelwood model (reaction between co-adsorbed Red molecule and oxygen) or Iley-Rideal model (reaction of gas-phase Red and adsorbed oxygen) [36,37]. Unlike the mechanisms of catalysis which have been the subject for careful kinetic studies, the interaction of reducing gases with chemisorbed oxygen at the sensors surfaces has hardly been experimentally proven. The ionosorption model, i.e., oxidation by ionosorbed oxygen species, has often been speculatively used to describe the sensing mechanisms of reducing gases [32,33,34,35,38,39,40]. The major issue is the lack of techniques for the observation of chemisorbed oxygen species, especially under in situ reaction conditions [25]. The response to an oxidizing gas (Ox) is attributed to the competitive chemisorption of target molecules (e.g., NO_2_, O_3_, Cl_2_) which are stronger electron acceptors than O_2_:Ox_(g)_ + *m e*^−^ → Ox^m−^_(ads)_(5)

This leads to trapping of more electrons in the surface states and a higher potential barrier at the MOS surface, resulting in decreased conductivity in the presence of oxidizing gases (Figure 2). The sensing mechanism related to chemisorption of oxidizing gases (Equation (5)) was experimentally proven by spectroscopy studies, e.g., NO_2_ adsorption on the MOS surfaces with the formation of NO_2_^−^ was observed by in situ Raman measurements [41] and diffuse reflectance infrared Fourier-transform (DRIFT) spectroscopy [42].

#### 1.2.3. The Oxygen Vacancy Model of Sensor Response

The oxygen vacancy model describes the sensor response of *n*-type MOS in terms of reversible shift of equilibria between bulk oxygen anions and oxygen vacancy formation (Equation (1)) depending on the ambient gas composition [29,30]. The reducing gas (Red) favors the formation of oxygen vacancies via the binding with lattice O^2−^ anions at the oxide surface and desorption of the oxidation product (RO_n_):Red_(g)_ + *n* O^2−^_(lat)_ → RO_n(g)_ + *n* V_O_^2−^ ↔ RO_n(g)_ + *n* V_O_^−^ + *n e*^−^ ↔ RO_n(g)_ + *n* V_O_ + 2*n e*^−^(6)

As a result, the concentration of donor states with loosely bound electrons increases (Figure 3), and the conductivity of *n*-type MOS rises. The reaction of target gas with lattice O^2−^ anions (Equation (6)) with simultaneous cleavage of metal-oxygen bonds is considered within the Mars-van Krevelen model of catalytic oxidation over metal oxide catalysts [36]. It has been accepted that the oxygen vacancy mechanism of sensor response takes place under oxygen-lean conditions, e.g., when the reducing gases (CO, hydrocarbons, H_2_) are detected in inert gases [43,44]. Such conditions are unfavorable for oxygen chemisorption at the MOS surface, and the ionosorption model of sensor response is unsuitable. Response to an oxidizing gas (Ox) is attributed to the fulfilment of an oxygen vacancy by an oxygen atom and restoration of lattice O^2−^ anions and metal-oxygen bonds:Ox_(g)_ + V_O_ + 2 *e*^−^ → R_(g)_ + O^2−^_(lat)_(7)
where the Ox molecule contains oxygen atoms and R is the reduction product. The participation of oxide anions in the interaction of MOS with analyte gases was confirmed experimentally in a number of works. The evolution of Sn^2+^ cations was observed by Mossbauer spectroscopy as a result of partial lattice oxygen reduction in SnO_2_ during the interaction with CO in inert atmosphere [45]. By DRIFT spectroscopy it was established that the W-O bonds were cleaved on the surface of WO_3_ in presence of CO in air [46]. On the contrary, the W-O bonds restored when WO_3_ was oxidized by NO_2_ or even H_2_O molecules in humid air [46,47]. Thus, the sensor response of tungsten oxide has been appropriately described by the oxygen vacancy model.

### 1.3. The Concerns on Sensor Materilals from Heterogeneous Catalysis

The reception of gas molecules at the sensor surface consists of same processes as on the surface of heterogeneous catalysts: adsorption and redox conversion of gas molecules. The correlations of sensitivity with catalytic activity of MOS have been observed experimentally [48,49,50]. Typical studies of heterogeneous catalysts consist of in-depth materials characterization, including the determination of active surface sites and the investigation of gas-solid interactions (adsorption, redox conversion) which are controlled by the surface sites [36,51]. On the contrary, in studies of sensors the greater attention has been paid to electrophysical phenomena (charge transfer, surface charging, band bending, conduction paths); and the sensitivity was successfully modelled depending on semiconductor type, donor states concentration, particle size, materials morphology, heterojunctions etc. [1,2,3,4,8,23,52,53]. Characterization of active sites at sensors surfaces was hardly performed, therefore, the chemical routes of analyte gas adsorption and redox interaction with the surface was incompletely understood. Catalytic activity of MOS and additives used in the composites is often referred to when gas sensing is discussed; however, the characterization of the surface sites and the direct experimental evidences for the catalytic interaction with gases was most often missing. The additives utilized in MOS-based sensors have traditionally been grouped as the catalytic (chemical) and the electronic ones [4,8,21]. The disadvantage of such classification is that the origin and mechanism of the catalytic activity in the process of target gas conversion is disregarded, and, consequently, the improvement of selectivity is not substantiated from the viewpoint of surface chemistry. Moreover, the additives (noble metals, metal oxides, carbon nanomaterials, polymers) can simultaneously exhibit the catalytic behavior and electronic interaction, i.e., heterojunction formation with MOS.

### 1.4. The Comparison of Energetic Parameters of n-Type Metal Oxide Semiconductors

In this review, we summarize our recent works aimed at reveling the active sites at the surface of gas sensitive MOS-based materials and the impact of surface sites on the adsorption and redox interactions with gas molecules which influence the sensing behavior. We show that the paradigm of heterogeneous catalysts research is worth applying for the investigation of sensor materials, since the reasons for selectivity in gas sensing may be fundamentally rationalized in terms of surface chemical reactivity. The approaches conventionally used for catalysts research were successfully transferred to the studies of gas sensors, including the qualitative and quantitative determination of active sites by surface science techniques such as temperature-programmed reduction and temperature-programmed desorption of probe molecules. Using in situ infrared spectroscopy the roles of adsorption sites and redox sites were revealed in the processes of gas molecules interaction with the sensors surfaces. The dependence of materials functional behavior on gas-solid interaction is the common point for heterogeneous catalysts and sensors. The sensitivity of sensors is much influenced also by the transduction of charge carriers between the surface and bulk of semiconductor and through the network of conduction paths. However, in the present review we propose that the selectivity of sensor response is mainly controlled by gas molecules reception, which in turn depends on active sites at the sensor surface. The nature and concentration of active sites depend on the chemical composition of sensing materials. Revealing these correlations will be useful for predictable control of sensors selectivity through appropriate tailoring the materials composition. Herein, the effect of chemical composition of simple *n*-type MOS (In_2_O_3_, ZnO, SnO_2_, TiO_2_, WO_3_) and mixed-metal oxides (BaSnO_3_, Bi_2_WO_6_) on the acid sites, oxidizing sites (surface oxygen species) and donor sites (oxygen vacancies) is discussed. The materials are wide-bandgap *n*-type semiconductors and differ in energetic parameters of the oxides and constituent cations (Table 1) [54,55]. Electronegativity of the cations increases in the order Zn^2+^, In^3+^, Sn^4+^, Ti^4+^, W^6+^; it agrees with the increment of charge/radius ratio. Accordingly, the metal-oxygen bond energy (*E*_M-O_) increases in respective metal oxides (Table 1) [54,56], and electron energy band positions (valence band top *E*_v_, conduction band bottom *E*_c_, middle of bandgap χ) decrease (Figure 4). The conduction band minima are mainly contributed by empty cationic s-orbitals (In^3+^, Zn^2+^, Sn^4+^) or d-orbitals (Ti^4+^, W^6+^), and the valence band maxima—by filled oxygen 2p-orbitals [57,58]. Energy levels of the orbitals shift deeper below the vacuum level as the electronegativity of cations increases (Figure 4). Bandgap width of oxides is related to cationic Pearson’s hardness (or, ionic-covalent parameter), which is close for the mentioned cations (intermediately hard) and, therefore, the oxides have wide bandgaps (*E*_g_ = 2.8–3.6 eV) [54]. In the following, we use M-O bond energy as the fundamental descriptor parameter of the active sites and gas sensitivity of metal oxides. The roles of acid sites, chemisorbed oxygen and oxygen vacancies in adsorption and redox conversion of gas molecules NH_3_, SO_2_, H_2_S, volatile organic compounds (VOCs), and NO_2_ is discussed in relation to sensing behavior. The key role of noble metal clusters (Pd, Ru) in specific catalytic oxidation of analyte molecules (CO, NH_3_) improving the selectivity of sensor response of catalytically functionalized MOS is discussed.

## 2. Types of Active Sites at the MOS Surface

In the literature there is no strict definition of an active site at the metal oxide surface; rather, it is conditional according to the materials and processes under consideration. In catalysts, the proper representation of an active site is the surface atom or ensemble which binds to a foreign molecule, thus loosening the intramolecular chemical bonds and forming the intermediates and/or final products of the catalytic reaction [36,64]. Yet, such defined active sites are hardly observable experimentally and unmeasurable due to their low fraction in the overall number of surface atoms, the specificity of catalytic reaction and the dynamic lability of active species in course of reaction. The other way demonstrated in some works was to regard the active sites as the adsorption sites for the specific reactant or probe molecule (e.g., isopropanol [65], aromatics [66,67]). Such defined active sites, i.e., adsorption sites, were quantified by the adsorption measurements and their concentration correlated with the catalytic activity; yet, no observations under in situ reaction conditions were available.

Herein, we use the conditional term of an active site as the surface species that possess a definite chemical behavior, like acidity/basicity, redox reactivity, electron donor/acceptor behavior, specific chemisorption capacity for probe molecules [64,65,66,67,68]. The observation of actual reactivity of surface sites under in situ reaction conditions is an unresolved problem; and the above definition refers to a potential participation of certain types of active sites in real-life chemical reactions that determine the functionality of catalytic and sensing materials.

In general, the active sites at a metal oxide surface can be constituted by coordinately unsaturated cations and oxygen anions, atomic ensembles, molecular and atomic adsorbates [69,70]. Defects are also considered as important active sites in catalytic reactions; including the point defects (cationic or anionic vacancies, atomic interstitials), and the extended defects like steps, corners, edges, dislocations, perimeter between metal oxide support and catalytic clusters [71,72,73]. The adsorption capacity and reactivity of surface sites is due to dangling bonds and unsaturated coordination of the peripheral atoms. The surface sites can form localized electron energy states and influence the band energy levels in MOS, thus influencing the electric response to surface processes. Coordinately unsaturated cations, oxygen anions and defects (vacancies, interstitials, dislocations, etc.) are the possible intrinsic active sites. Adsorbed oxygen, OH-groups, hydrogen adatoms resulting from spontaneous adsorption of oxygen and humidity represent the intrinsic adsorbate sites at the surface of pristine metal oxides. The extrinsic active sites can be created via the introduction of additives: atoms, atomic groups, clusters, nanoparticles of metals or metal oxides. In accordance with the chemical reactivity, the intrinsic active sites can be grouped into acid/base and reducing/oxidizing (or, donor/acceptor) ones [68]. The distinct chemical reactivity allows for the experimental determination of these species using appropriate techniques. In Figure 5 the types of active sites at the surface of tin oxide are shown schematically.

The concentration of different active sites depends on the nature of metal oxide (cationic charge and size, metal-oxygen bond energy), the synthesis conditions, microstructure, and additives. In the following sections, these relationships are reviewed based on our experimental studies of nanocrystalline *n*-type MOS: ZnO, In_2_O_3_, SnO_2_, BaSnO_3_, TiO_2_, WO_3_, Bi_2_WO_6_. The materials were synthesized via the standardized aqueous precipitation of metal hydroxides followed by calcination at temperature 300–700 °C [63,74,75,76,77,78].

**Figure 5 sensors-21-02554-f005:**
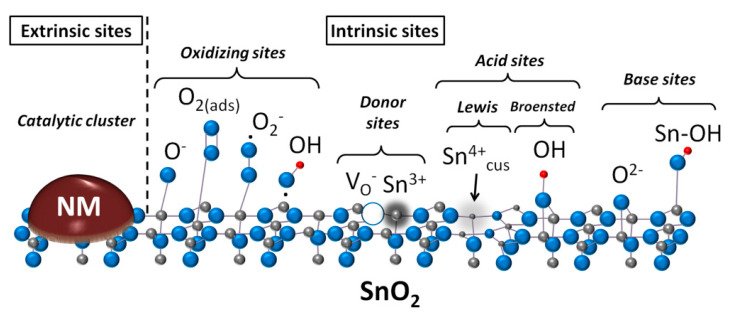
Types of active sites at the surface of tin oxide: noble metal cluster (NM), chemisorbed oxygen species (O_2(ads)_, O_2_^−^, O^−^), charged oxygen vacancies (V_O_^−^), partially reduced cations (Sn^3+^), coordinately unsaturated cations (Sn^4+^_cus_), hydroxyl species (OH, Sn-OH), surface oxygen anions (O^2−^). Adapted with permission from ref. [79]. Copyright 2014 American Chemical Society.

### 2.1. Acid/Base Sites

Surface metal cations are the Lewis acid sites, since the positive charge and unsaturated coordination favor the acceptation of lone electron pairs from the adsorbate to the cationic empty orbitals. From the electrostatic attraction law, Lewis acidity is expected to increase with increasing charge/radius ratio of the cation. On the contrary, the covalent contribution to cation-adsorbate bonding can be crucial for the acid-base bond strength. For example, it was shown that Sn^2+^ cations at the surface of partially reduced SnO_2_ are formally stronger Lewis acid sites than Sn^4+^ due to more covalent binding with adsorbed ammonia [80]. An alternative approach to quantitative scaling of the Lewis acidity is based on the concept of optical basicity (Λ); it is related to the ability of cations to shift electron density from oxygen anions in metal oxides [81].

Adsorption of H_2_O molecules (Lewis base) and strong binding to surface cation (Lewis acid) results in water molecule dissociation producing the OH-groups (Figure 6). In case of binding to strongly Lewis acidic cations and/or binding to more than one cation in a bridging conformation, the OH-group possesses an acidic proton and behaves as Brønsted acid site. Hydroxyl groups which are loosely bound to weakly acidic cations in a terminal conformation can behave as Brønsted base or acid sites depending on the counterpart interacting species. The OH-groups can be distinguished using FTIR spectroscopy: the terminal hydroxyls give rise to separate O-H peaks with wavenumbers above 3600 cm^−1^ [82,83]. The broad O-H bands centered at about 3400 cm^−1^ are indicative of the families of bridging hydroxyls associated via hydrogen bonds (Figure 7).

Surface oxygen anions O^2−^ are the intrinsic base sites at a metal oxide surface, and can act as the Lewis or Brønsted base depending on the adsorbate. The oxide anions can be determined by X-ray photoelectron spectroscopy (XPS). The asymmetric O 1s signal of nanocrystalline metal oxides can be simulated by two components (Figure 8): the major signal with binding energy 530.0–530.5 eV due to bulk O^2−^ anions and the lower peak of surface oxygen species at higher binding energy (531.0–533.0 eV) [34,85,86].

### 2.2. Oxidizing Sites

The oxidizing surface sites are most often associated with adsorbed oxygen [2,8,12,85,86,88,89]. There are several types of adsorbed oxygen species on a metal oxide surface:physisorbed O_2_ molecules held at the surface via van der Waals attraction, the physisoption dominates at temperature below −70 °C [90];chemisorbed O_2_ molecules covalently bound with surface cations via a local redistribution of electron density, the chemisorbed oxygen was found to dominate at the surface of tin oxide at temperature below 200 °C [91];ionosorbed species O_2_^−^, O^−^, and O^2−^ are formed by molecular and dissociative oxygen adsorption and acceptation of delocalized electrons from the bulk of MOS (Equation (2)) [5].

Besides, hydroxyl species were found to possess oxidizing activity to reducing gases: CO, H_2_ [82,83,92]. Metal cations in high oxidation states (Sn^4+^, Ti^4+^, V^5+^, Mo^6+^, W^6+^) represent the other type of oxidation sites at the surface of respective metal oxides. Of particular interest are the double M=O bonds which end up the metal-oxygen framework at the surface of V_2_O_5_, MoO_3_, WO_3_, and have been considered as the active oxidizing sites on these metal oxides and composites [93]. The reduction of surface metal cations to lower oxidation state proceeds through the cleavage of metal-oxygen bonds leaving oxygen vacancies in the surface structure of metal oxide. It is the so-called Mars-van Krevelen mechanism in heterogeneous oxidation catalysis.

### 2.3. Electron Donor Sites

Oxygen vacancies (V_O_^2−^, V_O_^−^) and reduced metal cations possess loosely bound electrons and behave like donor sites [4]. Oxygen vacancies which diffuse into the oxide bulk form donor states and account for the *n*-type conductivity of MOS: In_2_O_3_, ZnO, SnO_2_, TiO_2_, WO_3_ [22,58]. In these oxides, metals have the highest oxidation states for the respective elements group and, hence, can be reduced to a lower oxidation state by trapping electrons liberated from oxygen desorption and ionization of oxygen vacancies (Equation (1)).

## 3. Active Sites Concentrations at the Surface of Nanocrystalline *n*-Type MOS

In Table 2, we summarized the concentrations of active sites determined in our works on *n*-type simple MOS (ZnO, SnO_2_, TiO_2_, WO_3_) with different phase compositions and microstructure parameters, and mixed-metal oxides (BaSnO_3_, Bi_2_WO_6_). The simple MOS were synthesized by aqueous deposition of metal hydroxides with subsequent annealing at temperature 300–700 °C. Mixed-metal oxides were obtained by co-precipitation of metal hydroxides and hydrothermal treatment with subsequent annealing at 300–500 °C. The materials consisted of phase-pure nanocrystalline wurtzite-like ZnO, rutile-like SnO_2_, monoclinic γ-WO_3_, cubic perovskite-like BaSnO_3_, and Aurivillius phase Bi_2_WO_6_ with the crystallite size in the range *d*_XRD_ = 3–50 nm and specific surface area 4–100 m^2^/g (Table 2) [63,74,75,76,77,78]. Mixed-phase TiO_2_ with the rutile:anatase ratio of 3:2 was obtained after calcination at 700 °C [78]. TEM and SEM micrographs of the materials in Figure 9 and Figure 10 demonstrate the random shaped morphology of the nanoparticles of materials. Active sites of different types were determined by electron paramagnetic resonance (EPR) and probe molecules techniques: temperature programmed desorption of ammonia (TPD) and temperature programmed reduction by hydrogen (TPR). The principles and procedures of the active sites determination are described in the Appendix A.

### 3.1. Acid Sites

Acid sites were determined by TPD of ammonia [63,75,78,79,98]. Desorption patterns are shown in Figure 11a. The low-temperature bands at 50–200 °C were due to desorption of weakly bound probe molecules. Molecular NH_3_ evolved at this temperature, as was confirmed by mass-spectrometry (Figure 11b). These bands were ascribed to weak (Brønsted) acid sites, i.e., acidic OH-groups. At higher temperature (above 200 °C, Figure 11a) ammonia desorbed from stronger (Lewis) acid sites, i.e., coordinately unsaturated cations at the oxides surfaces. Concentration of Brønsted and Lewis acid sites increased in the order ZnO ≈ In_2_O_3_ < SnO_2_ < TiO_2_ < WO_3_ (Figure 12). It agrees with the increment of metal-oxygen bond energy, electronegativity and charge/radius ratio of cations (Table 1). In the same order does the optical basicity of oxides decrease [81]. The coincident trends of surface acidity, E_M-O_ and electronegativity should be due to increasing charge/size ratio for the cations Zn^2+^, In^3+^, Sn^4+^, Ti^4+^, W^6+^. The higher positive charge density at a cation (Lewis acid) favors stronger attraction of lone electron pair donor (Lewis base).

### 3.2. Donor Sites (Oxygen Vacancies)

Donor paramagnetic sites attributed to single charged oxygen vacancies V_O_^−^ were recognized on the EPR spectra of ZnO, SnO_2_, TiO_2_, WO_3_, by the signals with *g*-factor below 2.00 (Figure 13). The signal intensity was highly sensitive to temperature because of relaxation of excited V_O_^−^ spin states on lattice phonon vibrations [79]. Concentration of V_O_^−^ in MOS increases with increasing *E*_M-O_ (Figure 14a). The concentration was calculated per unit surface area (Table 2), although the sensitivity to phonon vibrations suggests that oxygen vacancies are mainly inside the oxide nanoparticles. Nevertheless, the values normalized per 1 mole of MOS followed the same tendency as in Figure 14a. Noteworthy, oxygen vacancies were detected in oxides under ambient conditions. The increasing concentration of V_O_^−^ in SnO_2_, TiO_2_ and WO_3_ agrees with the renowned oxygen deficiency of these compounds, e.g., the existence of Magneli phases (Ti_n_O_2n−1_) and numerous WO_3-x_ phases [99]. The stability of oxygen vacancies in MOS with high *E*_M-O_ can be explained by the relatively high electronegativity of the cations which can trap the loosely bound electrons, and, thus, be reduced to Sn^2+^ (and Sn^3+^ [100]), Ti^3+^, W^5+^ (and W^4+^), respectively. The formation of oxygen vacancy reduces the coordination number of cations, which is favorable for stronger (more covalent) bonding of electronegative cations with oxygen anions. On the other hand, the more stable are oxygen vacancies, the less readily should they donate electrons to chemisorbed acceptor molecules like O_2_. Therefore, the concentration of chemisorbed oxygen (including ionosorbates O_2_^−^) is lower on the surface of these oxides (Figure 14b). Oxidation of gas molecules (e.g., H_2_ in TPR) on the surface of oxides which are prone to oxygen deficiency is likely limited by the interaction with lattice oxygen anions (Equation (6)), i.e., described by Mars-van Krevelen mechanism in heterogeneous catalysis or oxygen vacancy model of sensor response.

### 3.3. Oxidizing Sites (Chemisorbed Oxygen)

The paramagnetic ionosorbates O_2_^−^ with the triplet EPR signal (*g*_1_ = 2.00, *g*_2_ = 2.01, *g*_3_ = 2.02–2.03 [25,103]) were observed on EPR spectra of nanocrystalline In_2_O_3_ [94,97], ZnO [102], SnO_2_ and TiO_2_ [96,101,104] (Figure 13). The concentration of O_2_^−^ at the oxides surfaces (10^−2^–10^−5^ micromole/m^2^) was by 2–6 orders of magnitude lower than that of oxidizing sites evaluated by TPR (Table 2). Hence, the active sites responsible for probe molecules oxidation were mainly constituted by other adsorbed and/or oxygen species and OH-groups, rather than O_2_^−^. The small concentration of O_2_^−^ is in line with Weisz limitation, which states that maximum coverage of ionosorbates at the semiconductor surface cannot exceed 10^−3^–10^−2^ monolayer, i.e., 1.7 × 10^−2^–1.7 × 10^−1^ micromole/m^2^ assuming that O_2_^−^ has normal orientation and radius close to r(O^−^) = 1.76 Å [8]. The concentration of O_2_^−^ determined by EPR decreased as the metal-oxygen bond energy increased (Figure 14b). As an assumption, from the oxides with lower *E*_M-O_ (ZnO, In_2_O_3_) lattice oxygen anions are desorbed into gas phase (Equation (1)). Thus formed oxygen vacancies are unstable due to low electronegativity of the cations In^3+^, Zn^2+^ and readily adsorb oxygen, including the formation of O_2_^−^ (Equation (2)). Noteworthy, the fraction of O_2_^−^ in oxidizing sites concentration was larger in In_2_O_3_: n(O_2_^−^):2n(H_2(TPR)_) ≈ 4 × 10^−2^. Thus, the donor sites (oxygen vacancies) at the surface of oxides with low *E*_M-O_ are mostly blocked by chemisorbed oxygen species from air; it limits the surface population by other adsorbates (e.g., neutral adsorbed O_2_, OH-groups) which could serve as the oxidizing surface sites.

Oxidizing surface sites were evaluated using TPR with hydrogen [63,75,76,78,79]. TPR profiles are shown in Figure 15. The oxides SnO_2_, In_2_O_3_ and WO_3_ were reduced to metals at temperature 400–900 °C. Temperature of reduction rates maxima (*T*_m_) shifted from *T*_m_ = 500–660 °C for In_2_O_3_ and SnO_2_ to *T*_m_ = 700–900 °C for WO_3_, which agrees with increasing thermodynamic stability of the oxides, as follows from increasing *E*_M-O_ and decreasing formation enthalpy (Table 1). The maxima of bulk reduction rates of SnO_2_ and WO_3_ also shifted to higher temperature with the increase of particle size. It can be due to higher thermodynamic stability with increasing crystallinity, as well as due slower reduction kinetics of larger MOS nanoparticles. TiO_2_ cannot be reduced completely by hydrogen, and the reduction of ZnO was not completed under the TPR conditions to prevent the evolution of Zn vapor. Low intense hydrogen consumption bands below 300 °C were attributed to the reduction of oxidizing surface sites like chemisorbed oxygen (O_2(ads)_, O_2_^−^, O^−^), surface oxygen anions (O^2−^) and hydroxyl species:O_n_^m−^_(surf)_ + 2*n* H_2(g)_ = 2*n* H_2_O_(g)_ + *m e*^−^_,_ (*n* = 1, 2; *m* = 0, 1, 2)
OH^−^_(surf)_ + 1/2 H_2(g)_ = H_2_O_(g)_ + *e*^−^(8)

Concentration of oxidizing surface sites n(H_2,TPR_) in Table 2 was calculated equivalent to micromoles of H_2_ consumed per 1 m^2^ of surface area. It was shown to be affected by the microstructure parameters. On the examples of SnO_2_ or TiO_2_ it can be seen that the increment of particle size and decrease of BET area results in the drop of n(H_2,TPR_) (Table 2). It can be assumed that smaller nanoparticles possess more defect-site atoms (at the edges, steps, corners, etc.) which hold the adsorbed oxygen and OH-groups more strongly (or, the smaller particles are more easily reducible) than regular atoms on a crystal surface, and the number of adsorbates per unit surface area increases. To estimate the effect of the metal oxide composition on the surface reducibility, the n(H_2(TPR)_) values were plotted in relation to metal-oxygen bond energies in Figure 14c. Noteworthy, the higher concentration of oxidizing sites was found for the oxides with intermediate *E*_M-O_ (SnO_2_, TiO_2_), in comparison to In_2_O_3_ (lower E_In-O_) and WO_3_ (higher E_W-O_). The origin of the volcano-shape dependence may be deduced from the trends of descending O_2_^−^ (Figure 14b) and growing V_O_^−^ concentrations with increasing *E*_M-O_ (Figure 14a). The enhanced surface reducibility of MOS with intermediate metal-oxygen bond energy might result from a proper balance between oxygen vacancy formation (Equation (1)) and oxygen chemisorption (Equations (2) and (3)). On the one hand, the oxides ZnO, SnO_2_, TiO_2_ possess oxygen vacancies which are persistent in ambient air (in contrast to those in In_2_O_3_) and may serve as adsorption sites for oxygen species (including O_2_^−^ observed by EPR) and OH-groups. On the other hand, oxygen vacancies associated with the cations Zn^2+^, Sn^4+^, Ti^4+^ should be rather unstable and adsorptive, in comparison to those associated by electronegative W^6+^ cations in oxygen-deficient WO_3_.

## 4. From Simple to Mixed-Metal Oxides: Metal-Oxygen Bond Energy and Active Sites

The complication of chemical composition of MOS is a powerful tool for tailoring the functional properties, including gas sensing behavior. The impurities may be introduced into the bulk of metal oxides or deposited onto the surface. Bulk additives can be grouped into dopants which do not alter the phase composition (e.g., Sb-doped SnO_2_ [105]; Sn-doped In_2_O_3_ [106]; Nb-doped TiO_2_ [107], etc.), and second components which condition the formation of new phases. Being incorporated into crystal lattice, dopants have minor effect on surface reactivity and are used for regulation of electronic properties of MOS. In this section, we focus on the examples of mixed-metal oxides which can be considered as derivatives of simple MOS, but exists in distinct crystalline phases: BaSnO_3_ in relation to SnO_2_, and Bi_2_WO_6_ in relation to WO_3_. The mixed-metal oxides were synthesized under hydrothermal conditions starting from freshly deposited SnO_2_∙xH_2_O and H_2_WO_4_. The former was mixed with Ba(OH)_2_ to obtain BaSnO_3_, and the latter—with Bi(OH)_3_ to obtain Bi_2_WO_6_ [63,77].

### 4.1. Crystal Structure and Metal-Oxygen Bonding

BaSnO_3_ has a perovskite-like cubic structure, which differs from the tetragonal rutile structure of SnO_2_ (Figure 16a). The structure of Aurivillius phase Bi_2_WO_6_ consists of alternating fluorite-like (Bi_2_O_2_)^2+^ and the perovskite-like (WO_4_)^2−^ layers. The latter are comprised by corner-shared octahedra {WO_6_}, which is relative to the structure of WO_3_ (Figure 16b). Semiconductor properties of the pairs of compounds BaSnO_3_–SnO_2_ and Bi_2_WO_6_–WO_3_ are similar: the simple and mixed-metal oxides are *n*-type semiconductors with close bandgap widths of *E*_g_ = 3.4–3.6 eV (BaSnO_3_, SnO_2_) [61,108] and *E*_g_ = 2.7–2.8 eV (Bi_2_WO_6_, WO_3_) [109,110]. Thus, the introduction of cations Ba^2+^ or Bi^3+^ and the accordant change of crystalline phases when the mixed-metal oxides are formed from SnO_2_ and WO_3_, respectively, do not affect semiconductor properties. The latter are determined by the framework of interconnected octahedra {SnO_6_} or {WO_6_}, which exist in the structures of simple oxides and mixed-metal oxides (Figure 16). Actually, DFT simulations showed that valence band maxima (VBM) in these compounds are mainly contributed by filled non-bonding O 2p-states [63,111,112]. The conduction band maximum (CBM) in SnO_2_ is mainly due to antibonding Sn 5s and O 2p states; and they are slightly contributed by Ba 6s states in the CBM of BaSnO_3_. The antibonding Sn 5s- and Ba 6s-orbitals are empty in both cases, which does not deteriorate metal-oxygen bonding in SnO_2_ and BaSnO_3_. Nevertheless, a slightly weaker Sn-O bonding in BaSnO_3_ can be deduced, respective to SnO_2_. Firstly, Sn-O distance is a little longer in BaSnO_3_ (2.06 Å [113]), respective to SnO_2_ (2.02 Å [61]). Secondly, from the TPR patterns (Figure 15) it can be inferred that bulk reduction of BaSnO_3_ started at a lower temperature (about 390 °C), than that of SnO_2_ (430 °C), judging by the data for materials with comparable particles sizes. Tin oxide was completely reduced in one step with the maximum rate of H_2_ consumption at *T*_m_ = 600–660 °C:SnO_2(s)_ + 2 H_2(g)_ = Sn_(l)_ + 2 H_2_O_(g)_(9)

Although BaSnO_3_ was incompletely reduced during TPR, the 1st hydrogen consumption peak centered at *T*_m_ = 500 °C was due to partial reduction of the oxide and, thus, corresponded to the cleavage of Sn-O bonds in the crystal lattice:BaSnO_3_(s) + ½ H_2(g)_ = ½ Ba_2_SnO_4(s)_ + ½ SnO_(s)_ + ½ H_2_O_(s)_(10)

The 2nd peak on the TPR pattern of BaSnO_3_ with the maximum at 890 °C was due to reduction of residual SnO.

In WO_3_ and Bi_2_WO_6_, the CBM is equally impacted by W 5d- and O 2p-states [63]. The empty Bi 6p-states do not contribute to CBM, but the filled Bi 6s states do affect the VMB. The Bi 6s–O 2p interaction is antibonding, which decreases the energetic stability of Bi_2_WO_6_ and diminishes W-O bond energy in the structure, respective to WO_3_. It follows from the comparison of integral partial crystal orbital Hamilton population (-IpCOHP) as a measure of bond energy, bond lengths and partial charges (Bader charge analysis) of atoms. Bi_2_WO_6_ possesses lower W-O bond energy (lower-IpCOHP), longer average W-O distance and increased ionicity of W-O bonds (higher partial atomic charges), in comparison to WO_3_ [63]. In agreement with the first-principles calculations, TPR data support the lower W-O bond stability in Bi_2_WO_6_ than in WO_3_ (Figure 15). Reduction of WO_3_ proceeded sequentially [63,75]:WO_3(s)_ + H_2(g)_ = WO_2(s)_ + H_2_O_(g)_, *T*_m_ = 700 °C(11)
WO_2(s)_ + 2 H_2(g)_ = W_(s)_ + 2 H_2_O_(g)_, *T*_m_ = 880 °C(12)

In Bi_2_WO_6_, bismuth was first reduced to metallic Bi at temperature *T*_m_ = 550 °C:Bi_2_WO_6(s)_ + 3 H_2(g)_ = 2 Bi_(l)_ + WO_3(s)_ + 3 H_2_O_(g)_(13)

The peaks of two-step reduction of residual WO_3_ were observed at *T*_m_ = 680 °C and *T*_m_ = 780 °C, i.e., at lower temperature than the respective ones for pristine WO_3_ (Figure 15). It indicates on the lower W-O bonds stability in the structure of Bi_2_WO_6_, in comparison to WO_3_ [63].

### 4.2. Concentration of Active Sites

The concentration of oxidizing surface sites estimated by hydrogen consumption in the lower temperature intervals (below 300 °C) on TPR patterns (Table 2) was close for SnO_2_ and BaSnO_3_, if samples with similar particle sizes were compared. The higher surface reducibility was observed for Bi_2_WO_6_, respective to WO_3_, which agrees with lower metal-oxygen bonds energy in the former.

Surface acidity of MOS is strongly influenced by the constituent cations. The concentration of Brønsted and Lewis acid sites evaluated by TPD was lower at the surfaces of BaSnO_3_ and Bi_2_WO_6_, in comparison to SnO_2_ and WO_3_, respectively (Table 2). It was shown that microstructure had minor impact on the difference of surface acidity between Bi_2_WO_6_ and WO_3_; it was mainly due to distinct chemical composition [63]. The lower acidity of mixed-metal oxides was attributed to the presence of Ba^2+^ or Bi^3+^ cations at the surfaces of BaSnO_3_ and Bi_2_WO_6_, provided that these cations have lower charge/radius ratio and, hence, lower electronegativity and weaker Lewis acidity than Sn^4+^ in pristine SnO_2_ or, moreover, W^6+^ in pristine WO_3_.

## 5. Impact of Active Sites on Gas Sensitivity of Nanocrystalline *n*-Type MOS

In this section, the correlations in active sites concentrations and the sensitivities of nanocrystalline *n*-type MOS to four analyte gases (NH_3_, CO, VOCs, NO_2_) are discussed based on our previously published research works. These correlations were deduced from the plots of sensitivity vs. metal-oxygen bond energy (Figure 17, Figure 18, Figure 19 and Figure 20) compared with the relations of active sites concentrations and *E*_M-O_ (Figure 12 and Figure 14). The specific cases of improved selectivity of mixed-metal oxides to SO_2_ and H_2_S are outlined.

The sensitivity (*S*) was defined as the relative change of resistance in presence of reducing (Red) or oxidizing (Ox) gases:*S*_Red_ = (*R*_air_−*R*_Red_)/*R*_Red_,
*S*_Ox_ = (*R*_Ox_−*R*_air_)/*R*_air_(14)
where *R*_air_ is sensor resistance in air, *R*_Red_—resistance in presence of reducing gases (NH_3_, CO, VOCs, SO_2_, H_2_S, etc.), *R*_Ox_—resistance in presence of oxidizing gas (NO_2_). It known that the sensitivity in general increases with the increment of specific surface area and the decrease of particle size [22,23,114], and we also observed it on the particular examples of nanocrystalline MOS and analyte gases [42,75]. In the syntheses of nanocrystalline MOS, the microstructural parameters could not be finely matched between different materials due to distinct energetics and kinetics of oxides crystallization. Therefore, in order to uniformly compare the sensitivities we defined an effective sensitivity (*S*_eff_) as the sensitivity normalized per 50 m^2^/g of the specific surface area of each sensor material [78]:*S*_eff_ = *S∙*50/SSA(15)
where SSA is the main value of the specific surface area of materials (Table 2). The value of 50 m^2^/g was chosen as an average expected SSA of nanocrystalline MOS obtained by the aqueous deposition or hydrothermal routes.

The effective sensitivities of MOS to the fixed concentration of analyte gases were compared in dry air conditions and measured at optimal operation temperature, i.e., at temperature of maximum sensitivity of the sensors to the given analyte gas. The large scattering of the *S*_eff_ data in Figure 17, Figure 18, Figure 19 and Figure 20 arouse from the sets of data compared, e.g., samples with same composition and different microstructure parameters, as well as errors in BET area evaluation.

### 5.1. Sensitivity to Ammonia and Surface Acidity

The sensitivity of different MOS to ammonia tends to increase for the MOS with increasing metal-oxygen bond energy (Figure 17a). Ammonia is a reducing gas, which is oxidized on the surface of pristine MOS mainly to N_2_ [31,115,116,117]:NH_3(ads)_ + 3/2*n* O_n_^m−^_(surf)_ = ½ N_2(g)_ + 3/2 H_2_O_(g)_ + 3*m*/2*n e*^−^(16)
although NO_x_ species were observed by mass-spectral analysis of desorbed gas in TPD of NH_3_ from NO_3_-contaminated In_2_O_3_ [78] or doped SnO_2_ [116].

**Figure 17 sensors-21-02554-f017:**
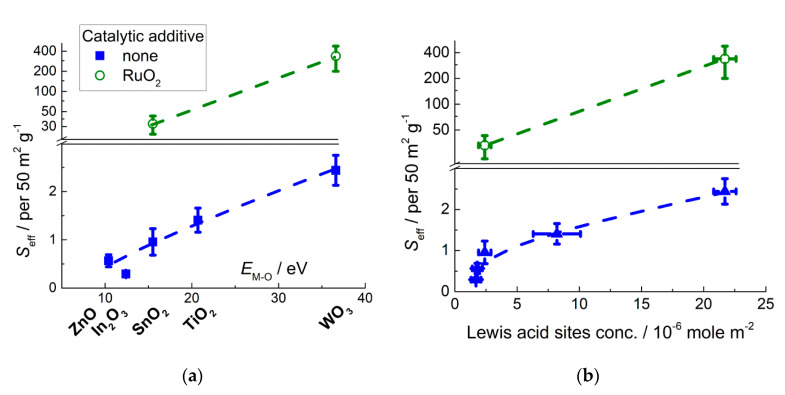
Sensitivity of pristine and RuO_2_-functionalized nanocrystalline *n*-type MOS to 20 ppm NH_3_ at temperature 150–250 °C in relation to metal-oxygen bond energy in MOS (**a**) and acid sites concentration at the MOS surfaces (**b**). Adapted with permissions from ref. [75] copyright 2018 Elsevier; ref. [98] copyright 2018 Elsevier; copyright [116] 2019 John Wiley and sons; ref. [117] copyright 2012 Elsevier.

DRIFT studies showed that basic NH_3_ molecules were chemisorbed on the acid sites and the binding with Lewis acid sites (surface cations) was strong enough, so that molecular adsorbates were predominant at the surfaces of MOS exposed to ammonia gas at elevated temperature [75,116,118]. Thus, the oxidation reaction (Equation (16)) which determines the sensor response is preceded by NH_3_ chemisorption on surface acid sites. The concentration of acid sites increases with the increment of metal-oxygen bond energy (Figure 12), which agrees with the increasing sensitivity to NH_3_ (Figure 17b). Thus, the sensitivity of MOS to NH_3_ is contributed by surface acidity that controls the chemisorption of analyte gas molecules.

### 5.2. Sensitivity to CO and VOCs Impacted by Oxidizing Sites and Acid Sites

The relation of CO sensitivity to metal-oxygen bond energy in MOS (Figure 18a) resembles that of oxidizing sites concentrations estimated by TPR (Figure 14c). Sensor response to CO is due its oxidation by surface oxygen and hydroxyl species at the surface of MOS [75,82,87]:CO_(g)_ + 1/*n* O_n_^m−^_(surf)_ = CO_2(g)_ + *m*/*n e*^−^(17)
CO_(g)_ + OH_(surf)_ = CO_2(g)_ + H^+^_(surf)_ + *e*^−^(18)

Besides CO_2_, carbonate species were detected by infrared spectroscopy on the surface of ZnO, In_2_O_3_ and SnO_2_ exposed to CO under certain conditions [5,76,119]. Carbon monoxide has no acid-base properties; hence, its adsorption is not influenced by surface acidity of metal oxides. Thus, it is reasonable that the sensitivity to CO is dependent on the surface reducibility of MOS which reflects the concentration of active species for CO oxidation (Equations (17) and (18)). In agreement with this is the increasing sensitivity with the concentration of oxidizing sites at the MOS surface (Figure 18b).

**Figure 18 sensors-21-02554-f018:**
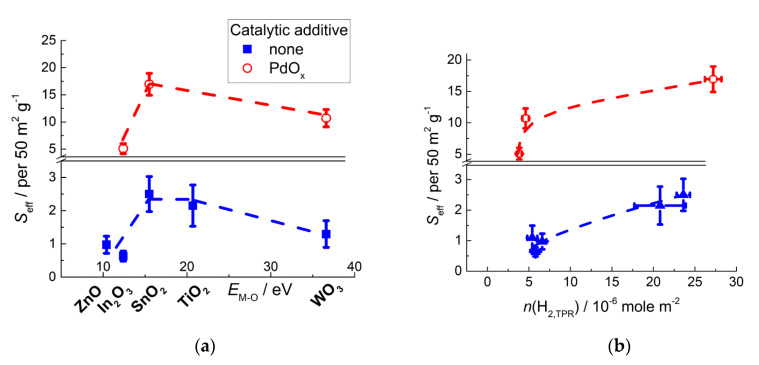
Sensitivity of pristine and PdO_x_-functionalized nanocrystalline *n*-type MOS to 20 ppm CO at temperature 250–300 °C in relation to metal-oxygen bond energy in MOS (**a**) and acid sites concentration at the MOS surfaces (**b**). Adapted with permissions from ref. [75] copyright 2018 Elsevier; ref. [76] copyright 2019 by the authors (BB BY); ref. [87] copyright 2010 Elsevier; ref. [97] copyright 2018 by the authors (CC BY); ref. [120] copyright 2015 by the authors (CC BY).

**Figure 19 sensors-21-02554-f019:**
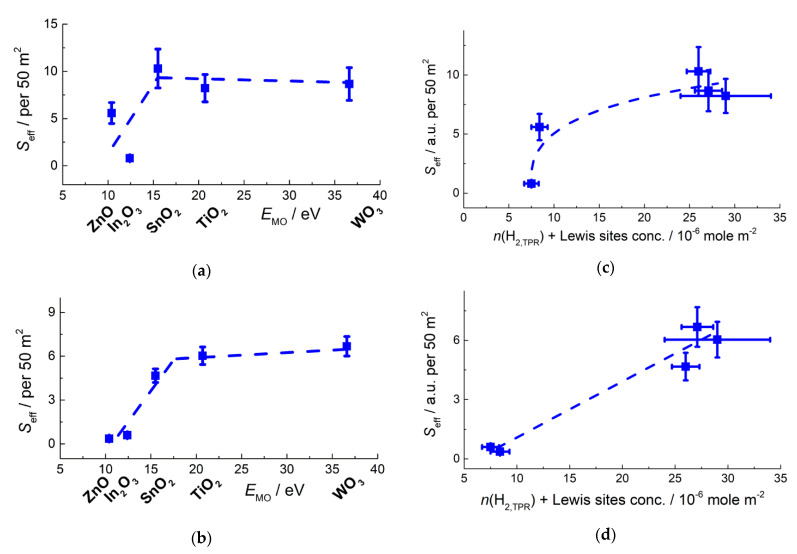
Sensitivity of pristine and Au-functionalized nanocrystalline *n*-type MOS to 20 ppm acetone (**a**,**c**) and 20 ppm methanol (**b**,**d**) at temperature 300 °C in relation to metal-oxygen bond energy in MOS (**a**,**b**) and sum of Lewis acid sites and oxidizing sites concentration at the MOS surfaces (**c**,**d**). Adapted with permission from ref. [78]. Copyright 2021 Elsevier.

The sensitivity to VOCs (methanol, acetone) increased with metal-oxygen bond energy and reached a plateau for SnO_2_, TiO_2_ and WO_3_ (Figure 19) [78]. On the one hand, it agrees with increasing concentration of oxidizing surface sites (Figure 14c). However, in contrast to the case of CO, the sensitivity of WO_3_ to VOCs did not drop, in spite of lower concentration of oxidizing sites. DRIFT study of MOS interaction with methanol and acetone vapors indicated that VOCs molecules adsorbed on surface cations and the adsorption enhanced with increasing surface acidity of MOS. It is likely due to Lewis basicity of oxygen atoms in CH_3_OH and CH_3_COCH_3_. At raised temperature that corresponds to sensors operation temperature, the adsorbed VOCs molecules were oxidized with the formation of carboxylate species [78,121]:CH_3_OH_(ads)_ + 5/2 O_n_^m−^_(surf)_ = HCOO^−^_(surf)_ + 3/2 H_2_O_(g)_ + 5*m*/2*n e^−^*(19)
CH_3_COCH_3(ads)_ + 9/2 O_n_^m−^_(surf)_ = CH_3_COO^−^_(surf)_ + CO_2(g)_ + 3/2 H_2_O + 9*m*/2*n e^−^*(20)

Thus, the sensitivity of MOS to oxygen-containing VOCs should be contributed by both processes: adsorption on Lewis acid sites and the adsorbates oxidation by oxidizing surface species (Equations (19) and (20)). The optimal sensitivity to VOCs could be expected for the oxides that have relatively high surface acidity balanced by surface reducibility, as it was observed for SnO_2_ and TiO_2_ (Figure 19). Then, the persistence of sensitivity in case of WO_3_ may be explained by its stronger surface acidity (Figure 12). The lower concentration of oxidizing sites at the surface of WO_3_ is compensated by the abundance of acid sites, which enforces the adsorption of VOCs and thus enhances the sensitivity. To illustrate this conclusion, the sensitivity to VOCs was plotted against the sum of Lewis acid sites and oxidizing sites n(H_2,TPR_) concentrations in Figure 19c,d.

### 5.3. Sensitivity to NO_2_ Determined by Donor Sites

The sensitivity to NO_2_ was found to enhance with increasing metal-oxygen bond energy of MOS (Figure 20), i.e., following the same trend as the sensitivity to NH_3_ (Figure 17). It is not an issue of selectivity, since the sensor resistance changes in opposite directions when exposed to oxidizing gas NO_2_ or reducing gas NH_3_. It has been established that sensing mechanism of NO_2_ relies on analyte molecules ionosorption with the formation of NO_2_^−^:NO_2(g)_ + *e*^−^ = NO_2_^−^_(ads)_(21)
which undergoes disproportionation and/or conversion resulting in NO_3_^−^, NO^+^, etc. [41,42,122]. Loosely bound electrons in *n*-type MOS are associated with oxygen vacancies (Equation (1)), the concentration of which (as suggested for V_O_^−^ by EPR) increases in MOS with increasing *E*_M-O_ (Figure 14a). Thus, the trend of increasing sensitivity to NO_2_ is due to higher concentration of electron donor sites in metal oxides with increasing *E*_M-O_.

**Figure 20 sensors-21-02554-f020:**
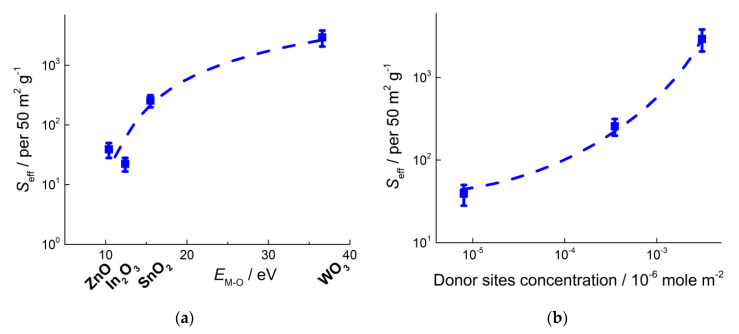
Sensitivity of nanocrystalline *n*-type MOS to 1 ppm NO_2_ at temperature 100–150 °C in relation to metal-oxygen bond energy (**a**) and donor sites concentration (**b**). Adapted with permissions from ref. [42] copyright 2019 by the authors (CC BY); ref. [63] copyright 2021 Elsevier; ref. [102] copyright 2013 Elsevier; ref. [123] copyright 2015 by the authors (CC BY).

### 5.4. Effect of Cations on Sensitivity and Selectivity of Mixed-Metal Oxides

The sensitivities of mixed-metal oxides and simple MOS to sets of various analyte gases are compared in Figure 21 for the pairs BaSnO_3_–SnO_2_ and Bi_2_WO_6_–WO_3_ [63,77]. Comparing the sensitivities of SnO_2_ to definite concentrations of analyte gases (Figure 21a), it may noticed that tin oxide was unselective in the detection of reducing gases and the difference in sensor signals might be attributed to different concentrations of analyte gases (e.g., higher sensitivity to 100 ppm H_2_ and lower sensitivities to 2–20 ppm CO, NH_3_, H_2_S, etc.). On the contrary, BaSnO_3_ demonstrated an improved sensitivity selectively to SO_2_ and, to a lesser extent, to alcohols; the sensitivity of BaSnO_3_ to other reducing gases was comparable with that of SnO_2_ (Figure 21a).

DRIFT spectroscopy revealed that the selectivity of BaSnO_3_ to SO_2_ was related with specific enhancement of target molecules oxidation due to binding the produced sulfate species with the Lewis basic sites (lattice O^2−^ anions) associated with Ba^2+^ cations [77]:SO_2(g)_ + 1/*n* O_n_^m−^_(surf)_ + Ba^2+^_(lat)_ + O^2−^_(lat)_ = BaSO_4(surf)_ + *m*/*n e*^−^(22)

Comparative sensing tests of Bi_2_WO_6_ and WO_3_ demonstrated the commonly higher sensitivity of bismuth tungstate to reducing gases [63]. The improved sensitivity of Bi_2_WO_6_ to VOCs, including formaldehyde and benzene, may be outlined (Figure 21b). Based on DFT simulations of W-O bonding in these structures, the higher sensitivity of Bi_2_WO_6_ to reducing gases was ascribed to the lower W-O bond strength and, hence, easier W-O cleavage on the surface during the oxidation of analyte gas molecules. On the other hand, the lack of sensitivity to NO_2_ was in line with the decreased concentration of donor sites (oxygen vacancies) in Bi_2_WO_6_, in comparison to WO_3_. It was evidenced by XPS evaluations that showed lower content of reduced W^5+^ cations in Bi_2_WO_6_ than in WO_3_ [63]. Also, the sensitivity of BaSnO_3_ to NO_2_ was lower than that of SnO_2_ (Figure 21a); it could be due to lower concentration of oxygen vacancies in the mixed-metal oxide. Indirectly, it was evidenced by the higher energy of oxygen vacancy V_O_ formation in BaSnO_3_ (above 4 eV [113]) than that in SnO_2_ (3.5 eV [124]). The other evidence for lower concentration of donor states in mixed-metal oxides is the higher baseline sensors resistance, in comparison to simple MOS [63,77]. Aside from the high sensitivity of WO_3_ to NO_2_, tungsten oxide displayed poor selectivity to reducing gases, as inferred from the comparable sensor signals in the range *S*_Red_ = 1–10 for the WO_3_ sensor (Figure 21b). On the contrary, Bi_2_WO_6_ displayed the selectively high sensitivity to H_2_S, which was also in the spotlight of other works [39,40]. It should be due to specific H_2_S-Bi^3+^ binding between the soft Pearson’s acid (bismuth cation) and base (sulfur atom in H_2_S).

## 6. Crucial Role of Catalytic Clusters in Sensitivity and Selectivity of Functionalized *n*-Type MOS

Surface functionalization by catalytic noble metal (NM) clusters (Figure 22) is a powerful approach to improve sensor characteristics of MOS. Well-documented advantages of catalytically modified MOS sensors include the enhanced sensitivity, decreased operation temperature, shortened response and recovery times, and improved selectivity to analyte gases [8,9,18,19]. Selectivity is a major issue in the sensing of reducing gases, since most of toxic, pyrogenic, flammable or explosive impurities in air belong to this group. The oxidizing pollutants are few; these molecules that have stronger electron affinity than oxygen, e.g., NO_2_, ozone, halogens. Therefore, the search for selective oxidation catalysts is challenging for the functionalization of MOS sensors to improve the sensing behavior in the detection of reducing gases. Based on the Sabatier principle, a proper oxidation catalyst enables the adsorption of reactants (reducing gas and oxygen) and intermediates, as well as desorption of reaction products [64]. The optimal activity and specificity is anticipated when the binding energies with the catalyst surface are balanced between these species [125]. Platinum group metals have moderate adsorption energies for various gases, since almost completed d-shell enables chemical bonding through electron acceptation from adsorbate as well as back-donation to the adsorbate [37]. Elements with less occupied d-orbitals as well as s-, p-, and f-elements easier donate electrons and too strongly adsorb oxygen (resulting in oxide formation) which deteriorates its oxidation activity. The adsorption capacity of elements with fully occupied d-orbitals (Ag, Au) is relatively low and is limited by the formation of electron vacancies in d-shell and nano-size effect [125,126,127].

The additives of NM form new (extrinsic) active sites at the surface of MOS (Figure 5), as well as influence its intrinsic active sites. The immobilized catalyst may impose the route of analyte gas conversion at the surface of MOS, which results in improved selectivity and sensitivity. The sensing mechanism in this case is dominated by the catalytic sites at the functionalized MOS surface [49]. In this section, we summarize our research on the effects of catalytic additives of PdO_x_, RuO_2_, and Au on the active sites and sensitivity of *n*-type MOS to CO, NH_3_, and VOCs. The concentration of noble metals was fixed at 1 wt.%, and the additives were introduced by impregnation of MOS with alcohol solutions of Pd(acac)_2_ or Ru(acac)_3_. The impregnated materials were annealed at minimum temperature required for the decomposition of acetylacetonates to maintain the dispersion of NM clusters and prevent aggregation [74,75,117]. Gold was introduced via the colloid adsorption technique in aqueous suspensions of MOS [78].

### 6.1. Role of PdO_x_ in Room-Temperature Sensitivity to CO

Palladium- and platinum-based catalysts have long been known as efficient catalysts for CO oxidation. The adsorption energy of oxygen on these metals (340–360 kJ/mole) is comparable to that of CO which is favorable for co-adsorption of reacting species [128]. The additives of Pd and Pt have often been recognized to improve sensing behavior of MOS to CO [33,75,87,129,130].

In our works, the nanocomposites MOS/PdO_x_ obtained after decomposition of deposited Pd(acac)_2_ consisted of agglomerated MOS nanocrystals covered by amorphous PdO_x_ clusters and nanoparticles with the size 1–15 nm (Figure 22a). Palladium was observed in mixed oxidation states PdO + Pd, with the dominance of the oxidized form [75,87,95]. Investigation of the active sites by surface science techniques suggested that PdO_x_ induced an increased concentration of aqueous species (adsorbed H_2_O, OH-groups, OH∙spin centers) at the surface of SnO_2_ [79]. However, no such effect was observed for WO_3_-based materials [75].

The notable effect of PdO_x_ clusters was the increased sensitivity of functionalized MOS sensors to CO (Figure 18). Valuably, the higher sensitivity was observed at room temperature [75,120]. Conventional MOS sensors require raised operation temperature (200–500 °C) [2,4,8,12], because of the activation barrier for analyte gas conversion at the oxide surface (Equations (4) and (6)). At room temperature the sensitivity of MOS-based sensors is usually vanished. Therefore, the outstanding room-temperature sensitivity of MOS/PdO_x_ can be utilized to improve the selectivity of CO detection and, by the way, to diminish the power consumption of the sensor device. DRIFT studies revealed that CO was chemisorbed specifically on Pd-sites at room temperature (Figure 23). It was deduced that Pd-CO binding loosened the interatomic bond in the adsorbate and thus catalyzed its oxidation by surface oxygen and hydroxyl species [75,118]:Pd-CO_(ads)_ + 1/*n* O_n_^m−^_(surf)_ = Pd_(surf)_ + CO_2(g)_ + *m*/*n e*^−^,Pd-CO_(ads)_ + OH_(surf)_ = Pd_(surf)_ + CO_2(g)_ + H^+^_(surf)_ + *e*(23)

Raising temperature suppressed CO chemisorption, which explained the drop of sensitivity to CO with increasing temperature [74,75,120].

### 6.2. Impact of RuO_2_ on Sensitivity and Selectivity to NH_3_

Ruthenium in the metallic Ru or oxidized RuO_2_ state is an efficient active phase of catalysts for selective processing of ammonia: synthesis, decomposition or oxidation [125,131,132]. DFT simulations suggested that due to moderate Ru-N bond energy, the chemisorption of NH_3_ is not too strong and its conversion to NH_x_-intermediates and final products (N_2_, NO_x_) proceeds with low activation barriers [125,133]. The catalytic effect of ruthenium in the oxidation of ammonia was found promising for the improvement of sensitivity and selectivity of Ru-modified MOS sensors [134,135,136].

In the nanocomposites MOS/RuO_2_ obtained via thermal decomposition of MOS-impregnated Ru(acac)_3_ at 265 °C, the additive was observed in the form of randomly shaped clusters and nanoparticles with the size ranging from 2–20 nm (Figure 22b) to few hundred nm [75,95]. Chemical state analysis indicated on the predominantly RuO_2_ state of ruthenium with a fraction of Ru^3+^ [74,75,95,117]. Agreeing results were obtained by TPR of SnO_2_- and WO_3_-based materials that the concentration of oxidizing sites increased at the surface of Ru-modified MOS [75,79].

Ru-modified MOS demonstrated substantially improved sensitivity and selectivity to NH_3_ at raised operation temperature (150–250 °C) [75,117,135,136]. It was demonstrated that SnO_2_/RuO_2_ was capable to discriminate NH_3_ traces in presence of interfering CO gas in air [120]. Comparing different ruthenated metal oxides, an increasing sensitivity to NH_3_ was found with increasing *E*_M-O_ and surface acidity of MOS (Figure 17). Similarly to the case of pristine MOS, this trend is explained by the increasing adsorption of NH_3_ on Lewis acid sites at the oxides surfaces. By DRIFT spectroscopy, the specific route of NH_3_ oxidation catalyzed by RuO_2_ was evidenced [75,118]. Although NH_3_ adsorption was almost unaffected by the NM additives, it was only on the Ru-modified MOS that Ru-bound NO species evolved from the interaction with ammonia (Figure 24) [75,118,136]. Nitrosyl species were further oxidized to NO_2_ gas [117]. Thus, the reason for improved sensitivity and selectivity to NH_3_ is that supported RuO_2_ nanoparticles specifically catalyze deep oxidation of adsorbed ammonia at the sensors surfaces, while in the absence of ruthenium the oxidation of ammonia likely stopped at the stage of N_2_ formation:NH_3(ads)_ + 5/2*n* O_n_^m−^_(surf)_ + Ru^4+^_(surf)_ = Ru^4+^-NO_(ads)_ + 3/2 H_2_O_(g)_ + 5*m*/2*n e*^−^,Ru^4+^-NO_(ads)_ + 1/*n* O_n_^m−^_(surf) =_ NO_2(g)_ + *m*/*n e*^−^(24)

The distinction between the surface sites of NH_3_ adsorption (acid sites) and oxidation (catalytic RuO_2_ sites) suggested an idea of independent co-functionalization of MOS surface by acidic and catalytic species for tailoring the sensor characteristics. For this purpose, SnO_2_ was loaded by different amounts of sulfate SO_4_-groups and RuO_2_ clusters [98]. It was demonstrated that introducing the comparable amounts (1–3 wt.%) of the acidic SO_4_- and catalytic RuO_2_ additives properly enhanced the adsorption of NH_3_ balanced by its oxidation, resulting in the optimum sensitivity.

### 6.3. Au-Promoted Oxidizing Sites and Sensitivity to VOCs

Gold nanoparticles supported on metal oxides have been in focus of extensive research in catalysis [37,72,126]. Although Au itself is inactive in the bulk state, it was reported that the catalytic activity of nanosized gold exceeded that of platinum group metals in low-temperature CO oxidation [126]. Gold nano-catalysts were also shown to be efficient for total oxidation of VOCs [137,138]. Functionalization of MOS sensors by Au resulted in improved sensitivity to CO [139] and VOCs [139,140], which was also attributed to catalytic activity of Au, electronic cluster-support interaction and spillover of oxygen. Catalytic activity of Au strongly depends on cluster size and on the support. Reducible metal oxides, including MOS, are the proper supports enabling the adhesion of Au clusters and active oxygen species for the catalytic reaction cycle [141]. Among the various metal oxides, TiO_2_-supported Au catalysts have attracted the great interest [72,92,126,142,143]. The higher extent of surface oxygen activation on TiO_2_/Au was ascribed to the proper Ti-O bond energy favorable for facile interplay between oxygen vacancy formation and oxygen chemisorption from gas phase [144]. However, the origin of active sites in oxide-supported Au catalysts has still been questionable: periphery Au and O atoms, charged Au+ species, Au-OH groups, etc., were suggested [92,126,145].

Functionalization of MOS surfaces by small gold clusters is challenging, since the mobility of Au atoms induces the aggregation of gold in larger nanoparticles at raised operation temperature of sensors. Furthermore, chloride ions evolved from the precursor HAuCl_4_ decomposition promote the aggregation of gold. To minimize it, the colloid adsorption technique of MOS modification by freshly deposited Au(OH)_3_ is preferable, rather than the impregnation by HAuCl_4_ [126]. Recently, we reported on a comparative study of Au-modified MOS sensors obtained by colloid adsorption and thermal decomposition of Au(OH)_3_ at above 200 °C [78]. The additive was observed in the metallic state Au^0^ in the form of nanoparticles with the size 5–30 nm. Investigation of active sites showed that in presence of Au the concentration of oxidizing sites increased at the surfaces of *n*-type MOS, and the relation *n*(H_2,TPR_ –*E*_M-O_ had the volcano shape with the maximum for TiO_2_/Au (Figure 25a), similarly to that for pristine MOS (Figure 14c).

The Au-functionalized sensors displayed higher sensitivity to methanol and acetone, respective to pristine MOS (Figure 25b,c). Noteworthy, the dependence of VOCs sensitivity on E_M-O_ in MOS was similar to that of oxidizing sites concentration (Figure 25a), i.e., with the maximum for TiO_2_/Au. On the other hand, it was dissimilar to the relation VOCs sensitivity–*E*_M-O_ for pristine MOS which had a plateau for SnO_2_, TiO_2_, and WO_3_ (Figure 19a,b). The plateau could be ascribed to increased adsorption of VOCs at the abundant acid sites of WO_3_ which compensated the lower oxidizing sites concentration at its surface. The sensitivity of Au-functionalized MOS to VOCs was less dependent on surface acidity and to a larger extent determined by the concentration of oxidizing sites (Figure 25d,e). DRIFT measurements showed no significant changes in the oxidation routes of CH_3_OH and CH_3_COCH_3_ on Au-functionalized MOS, respective to pristine oxides (Equations (19) and (20)) [78]. Thus, in presence of Au the oxidation of VOCs by surface oxygen was enhanced leading to higher sensitivity, but the sensing mechanism was not altered. The volcano-shaped relation with the maximum for TiO_2_/Au allows assuming that the oxidation reactions (Equations (19) and (20)) influence the sensitivity of MOS/Au to a larger extent, than the adsorption on the acid sites of MOS. It is in line with the concept of Au-promoted surface oxygen activation, which was most noticeable on titania-supported catalysts [144].

## 7. Conclusions

In this review of the authors’ research on sensor materials based on nanocrystalline *n*-type MOS (In_2_O_3_, ZnO, SnO_2_, TiO_2_, WO_3_) with particle sizes in the 3–50 nm range, the direct correlations between material composition, concentrations of active sites at the surface, and gas sensing behavior was substantiated. The active sites were classified as acid sites of Brønsted type (acidic OH-groups) and Lewis type (coordinately unsaturated cations), oxidizing sites (adsorbed oxygen, OH-species, and/or surface oxygen anions), and donor sites (oxygen vacancies and/or reduced cations). In line with the increment of charge/radius ratio and electronegativity of the constituent cations, the metal-oxygen bond energy reported in literature and surface acidity probed by TPD of ammonia increase in the order: In_2_O_3_, ZnO, SnO_2_, TiO_2_, WO_3_. Concentration of oxidizing sites evaluated by TPR was higher at the surface of the oxides with moderate *E*_M-O_ (SnO_2_, TiO_2_). Probably, it was due to interplay of two opposite trends: enhancing oxygen vacancies formation and decreasing oxygen chemisorption with the increase of *E*_M-O_, as was shown by the concentrations of paramagnetic sites V_O_^−^ and O_2_^−^ measured by EPR. The extrinsic catalytically active sites were formed via the functionalization of MOS surfaces by PdO_x_ and RuO_2_ clusters with the size 1–20 nm, and Au nanoparticles with the size 5–30 nm surfaces. Mixed-metal oxides BaSnO_3_ and Bi_2_WO_6_ obtained by hydrothermal synthesis were considered in comparison with SnO_2_ and WO_3_, respectively. In the pairs BaSnO_3_–SnO_2_ and Bi_2_WO_6_–WO_3_ there are similar structural units and semiconductor properties determined by the Sn-O and W-O networks, respectively. The introduction of Ba^2+^ slightly elongated Sn-O bond in BaSnO_3_, respective to SnO_2_. First-principles analysis showed that the filled 6s-level of Bi^3+^ deteriorates the stability of crystal structure and W-O bonding in Bi_2_WO_6_, in comparison to WO_3_. Due to the relatively low charge/radius ratio and electronegativity of Ba^2+^ and Bi^3+^ cations, the concertation of acid sites was lower at the surfaces of mixed-metal oxides, compared with SnO_2_ and WO_3_. Due to lower W-O bond energy, Bi_2_WO_6_ had higher oxidizing surface reactivity than WO_3_.

The trends in active sites concentration and sensitivity to different analyte gases (NH_3_, NO_2_, CO, VOCs) were compared for MOS using metal-oxygen bond energy as a descriptor parameter. It was shown that sensitivity of pristine and RuO_2_-modified MOS to NH_3_ increased with surface acidity reaching maximum for WO_3_-based sensors. Diffuse-reflectance infrared spectroscopy suggested that it was due to enhanced adsorption of NH_3_ on Lewis acid sites that favors the oxidation of adsorbed molecules and sensor response formation. The sensitivity to CO which has no acid-base behavior is likely determined by the oxidation reaction at the surface. The sensitivity of pristine and PdO_x_-modified MOS to CO correlated with the concentration of oxidizing sites which was higher for SnO_2_ and TiO_2_. The sensitivity to VOCs molecules possessing Lewis-basic O atoms (methanol, acetone) was contributed by the processes of adsorption on acid sites and the conversion on oxidizing sites. For pristine MOS, the sensitivity to VOCs reached a plateau for SnO_2_, TiO_2_, WO_3_. The higher surface acidity of WO_3_ and enhanced adsorption of VOCs could compensate its lower oxidizing sites concentration, respective to SnO_2_ and TiO_2_. The sensitivity to oxidizing gas NO_2_ increased with the increment of donor sites concentration (oxygen vacancies) in MOS with the maximum for pristine WO_3_.

Changing the composition of sensor materials modifies the metal-oxygen bonding and intrinsic active sites, as well as results in the formation of specific active sites and improved selectivity. Mixed-metal oxides BaSnO_3_ and Bi_2_WO_6_ possessed lower concentration of donor sites and decreased sensitivity to NO_2_, respective to SnO_2_ and WO_3_. The presence of Ba^2+^ cations at the surface of BaSnO_3_ favored the oxidation of sulfur dioxide to sulfate-species, which provided the selectively high sensitivity to SO_2_. The lower W-O bond energy and higher surface reducibility of Bi_2_WO_6_ correlated with its improved sensitivity to reducing gases including VOCs, in comparison to WO_3_. The selectivity of Bi_2_WO_6_ to H_2_S should be due to specific binding of hydrogen sulfide to Bi^3+^ cations.

Clusters and nanoparticles of noble metals (or noble metal oxides) with specific catalytic activity in the oxidation of certain gas molecules can significantly change the sensing mechanism. Pd sites specifically bind CO which imposes the sensitivity and selectivity of Pd-functionalized MOS to carbon monoxide at room temperature. Moreover, the lack of target molecules adsorption at raised temperature determines the drop of MOS/PdO_x_ sensitivity to CO. RuO_2_ clusters specifically catalyze oxidation of NH_3_ adsorbed at the surface of MOS producing NO_x_. It leads to a pronounced improvement in the selectivity to ammonia for RuO_2_-modified sensors, and the sensitivity increases with the surface acidity of MOS reaching maximum for WO_3_/RuO_2_. Gold nanoparticles promote oxidizing sites at the surface of MOS, likely due to activation of surface oxygen species. It results in improved sensitivity to reducing VOCs molecules. Although the route of methanol and acetone conversion did not change in presence of Au, the sensitivity of Au-functionalized sensors was mainly determined by the oxidation reaction and was higher for the SnO_2_/Au and TiO_2_/Au nanocomposites.

Further research is necessary to improve fundamental understanding of the interrelation “materials composition—active sites at the surface—sensing behavior”. We believe that surface science techniques like TPR, TPD, infrared spectroscopy (FTIR, DRIFT) are worth being included into the portfolio of characterization tools for sensing materials. Finding out the correlations between sensitivities to analyte gases and concentrations of adsorptive (acid/base) and reactive (redox) surface sites is informative on the processes which determine the sensitivity and selectivity, and DRIFT spectroscopy sheds the light on sensing mechanisms. The possible directions of future research on the active sites and gas sensitivity of nanocrystalline MOS include but are not limited to the following ones. (i) Design of new sensor materials based on simple MOS, mixed-metal oxides, and composites possessing specific active sites at the surface. The parameters that could be varied are the chemical composition, crystal structure and facets, morphology, concentration of additives. Experimental approaches for creating and controlling the concentrations of acid sites, reactive surface oxygen and oxygen vacancies should be developed, e.g., thermal quenching, chemical etching, surface modification, etc. It is challenging to stabilize the active sites in air under heating, so as to maintain the sensors stability during long-term operation. (ii) Extending the research of mixed-metal oxides, evaluation of the effects of cationic composition on metal-oxygen bonding, active sites, semiconductor properties and selectivity of gas—solid interaction. (iii) The search for new catalytic additives with specific reactivity towards redox conversion of definite analyte gases is challenging for theoretical and experimental studies, since the catalytic functionalization of MOS sensors is a powerful approach to improve selectivity and specify the temperature regime of gas sensing. Besides the conventionally used additives of noble metals and transition metal oxides, the potentially efficient catalysts may be found among bimetallic clusters, organically hybridized complexes, metal-organic frameworks, chalcogenides, carbon-based compounds, etc. The main issue would be stability of the catalytic additives under sensors operation conditions. Metal-oxygen bonding and active sites at the surface of MOS influence the cluster/support interplay which conditions the successive catalytic functionalization of sensors. Hence, different MOS should be comparatively investigated as the gas sensitive composites with a certain catalyst. (iv) Development of in situ and *operando* techniques for the revelation of kinetics and energetics of gas molecules interaction which with the active sites at the MOS sensors surfaces.

## Figures and Tables

**Figure 1 sensors-21-02554-f001:**
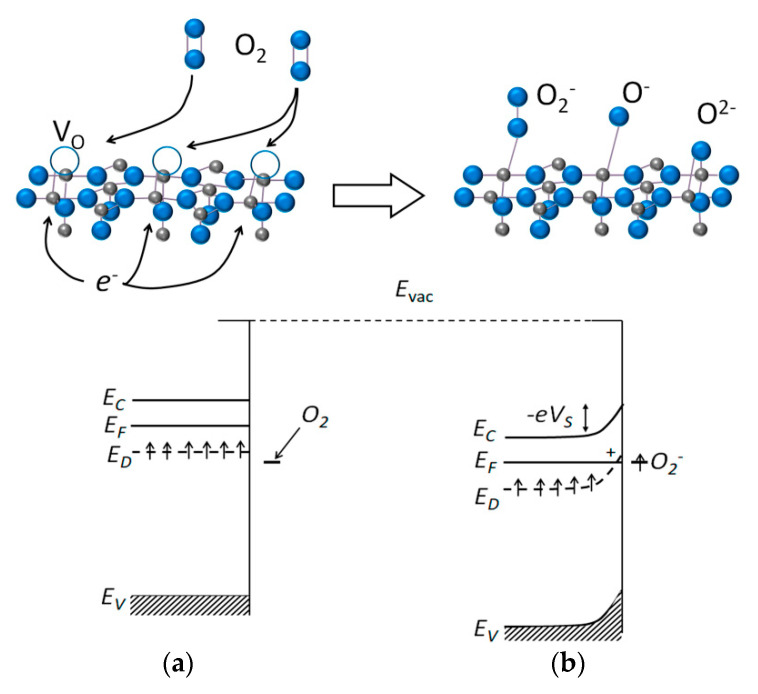
Schematic of representation (**top**) of the surface of MOS possessing oxygen vacancies (V_O_) before (**a**) and after oxygen ionosorption (**b**); and the corresponding modulation of band energy levels (**bottom**): vacuum level (*E*_vac_), conduction band bottom (*E_C_*), Fermi level (*E_F_*), donor states level (*E_D_*), valence band top (*E_V_*), potential energy surface barrier (−*eV*_S_).

**Figure 2 sensors-21-02554-f002:**
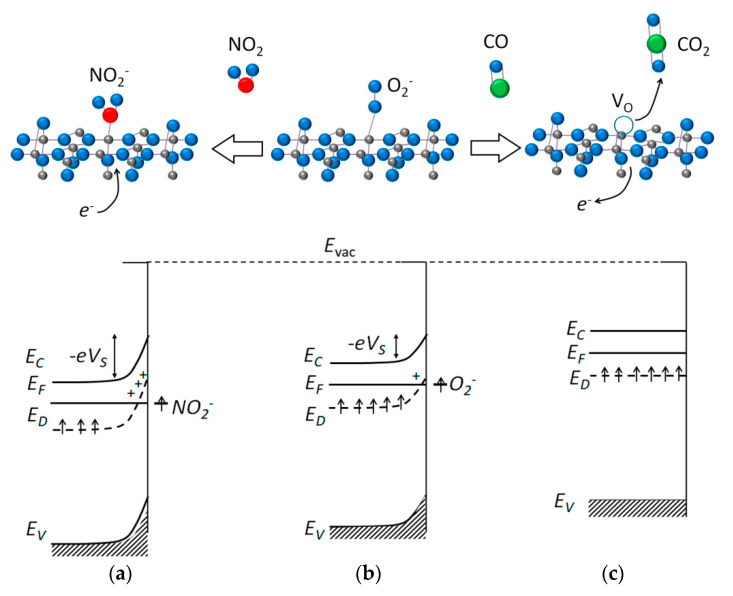
Schematic representation of the surface of *n*-type MOS with ionosorbed oxygen species (**b**) and after its interaction with reducing gas CO (**c**) and oxidizing gas NO_2_ (**a**) within the chemisorption model of sensor response (**top**); and the corresponding modulation of band energy levels (**bottom**).

**Figure 3 sensors-21-02554-f003:**
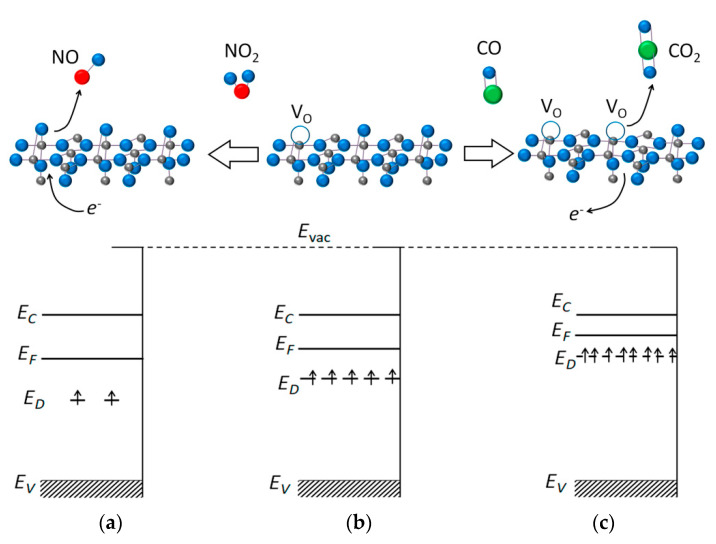
Schematic representation of the *n*-type MOS with oxygen vacancies and lattice oxygen anions (**b**), and after its interaction with reducing gas CO (**c**) and oxidizing gas NO_2_ (**a**) within the oxygen vacancy model of sensor response (**top**); and the assumed changes in donor states population (*E*_D_) and Fermi level positions (*E*_F_) (**bottom**).

**Figure 4 sensors-21-02554-f004:**
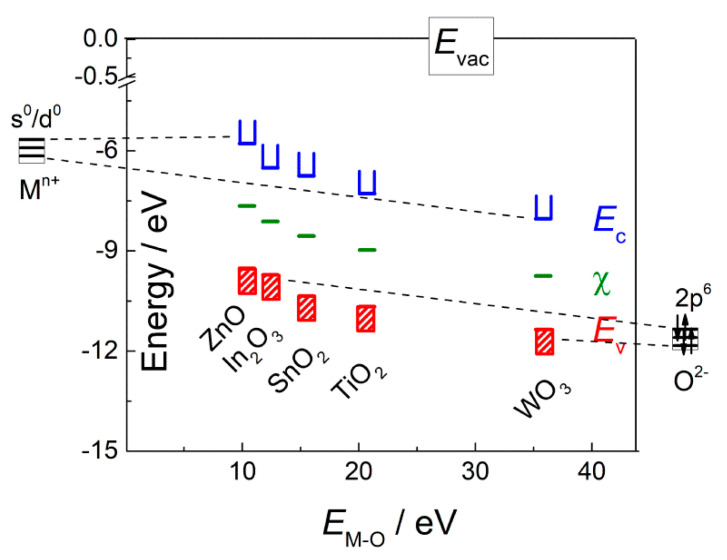
Positions of conduction band minima (*E*_c_), valence band maxima (*E*_v_) and electronegativity (χ) of *n*-type MOS, respective to vacuum level (*E*_vac_). Adapted with permission using numeric data from ref. [54]. Copyright 2011 Elsevier. The oxides electronegativity corresponds to middle bandgap position. The levels of atomic orbitals for metal cations (M^n+^ s^0^/d^0^) and oxygen anions (O^2−^ 2p^6^) are shown schematically.

**Figure 6 sensors-21-02554-f006:**
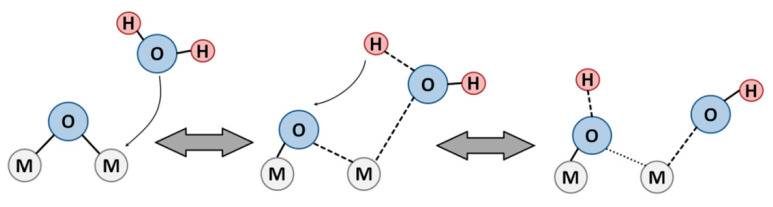
Scheme of water molecule dissociative adsorption at metal oxide surface with the formation of bridging and terminal OH-groups. Reprinted with permission from ref. [84]. Copyright 2018 Springer.

**Figure 7 sensors-21-02554-f007:**
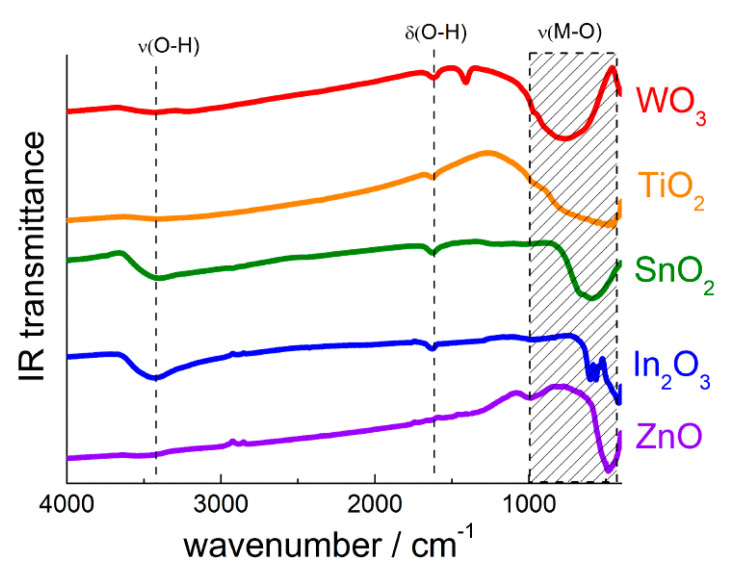
FTIR spectra of nanocrystalline *n*-type MOS. Adapted with permission from ref. [75] copyright 2018 Elsevier; ref. [76] copyright 2019 Marikutsa, Rumyantseva, Gaskov, Batuk, Hadermann, Sarmadian, Saniz, Partoens and Lamoen Creative Commons Attribution License (CC BY); ref. [78] copyright 2021 Elsevier; ref. [79] copyright 2014 American Chemical Society.

**Figure 8 sensors-21-02554-f008:**
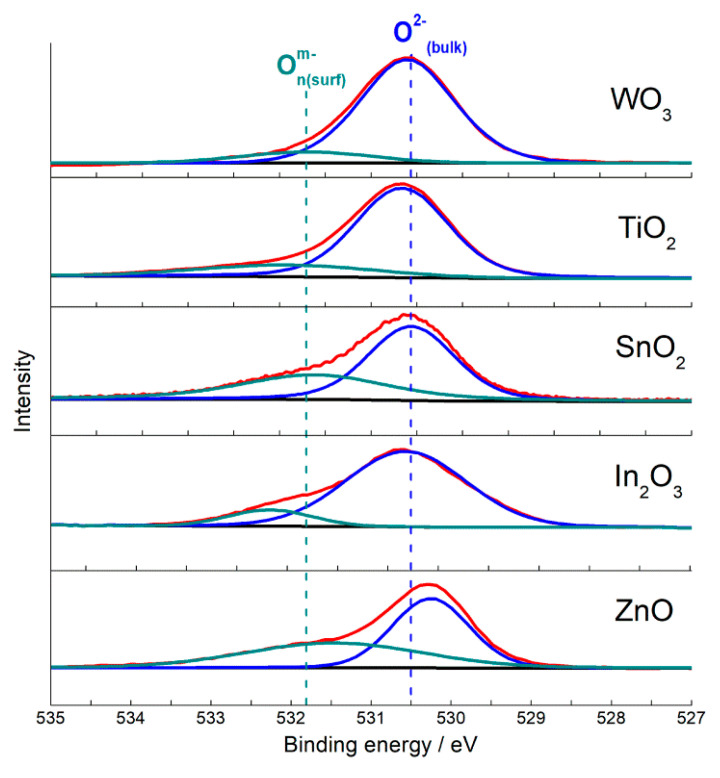
XP-spectra of O 1s state for nanocrystalline *n*-type MOS. Adapted with permissions from ref. [75] copyright 2018 Elsevier; ref. [78] copyright 2021 Elsevier; [79] copyright 2014 American Chemical Society; ref. [87] copyright 2010 Elsevier.

**Figure 9 sensors-21-02554-f009:**
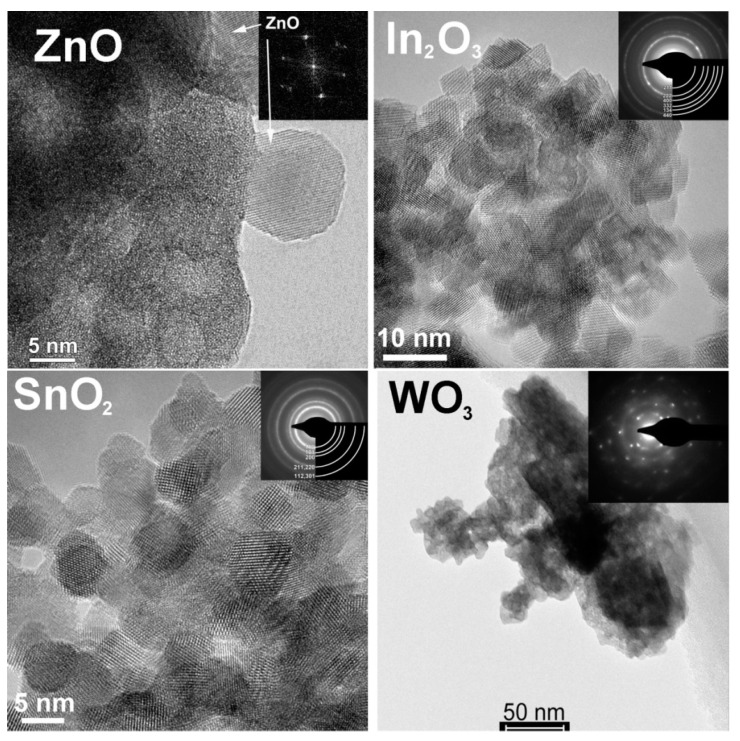
HRTEM images of nanocrystalline ZnO, In_2_O_3_, and SnO_2_, and TEM image of WO_3_; the sample were annealed at 300 °C. The insets show electron diffraction patterns. Adapted with permissions from ref. [42] copyright 2019 by the authors (CC BY); ref. [76] copyright 2019 by the authors (CC BY); ref. [94] copyright 2015 Elsevier; ref. [95] copyright 2013 American Chemical Society.

**Figure 10 sensors-21-02554-f010:**
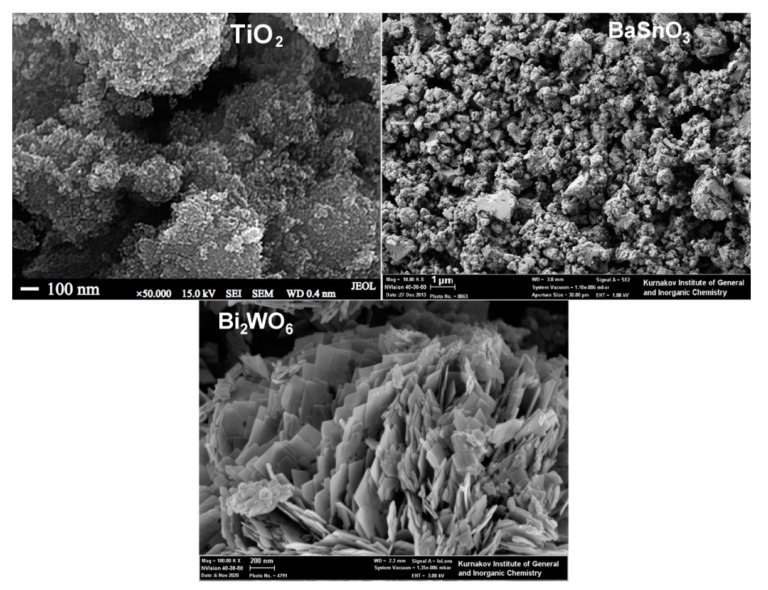
SEM images of nanocrystalline TiO_2_, BaSnO_3_, and Bi_2_WO_6_. Adapted with permissions from ref. [63] copyright 2021 Elsevier; ref. [77] copyright 2015 by the authors (CC BY); ref. [96] copyright 2021 Springer Nature.

**Figure 11 sensors-21-02554-f011:**
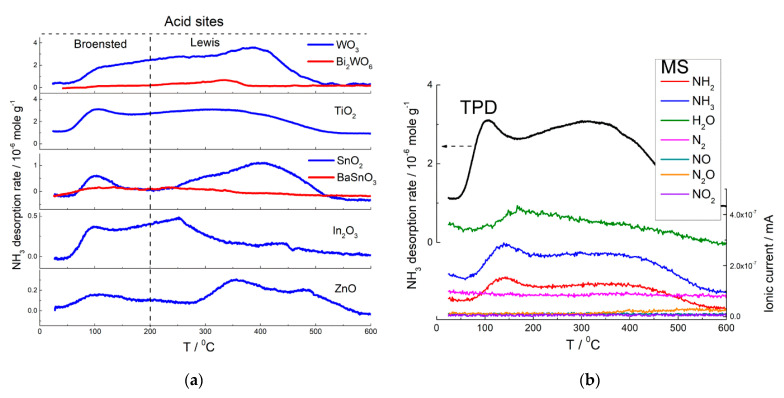
Temperature plots of ammonia desorption rate from the surface of *n*-type MOS (**a**). Adapted with permissions from ref. [63] copyright 2021 Elsevier; ref. [75] copyright 2018 Elsevier; ref. [78] copyright 2021 Elsevier; ref. [79] copyright 2014 American Chemical Society. TPD pattern of TiO_2_ compared with mass-spectral (MS) analysis of desorbed gas (**b**).

**Figure 12 sensors-21-02554-f012:**
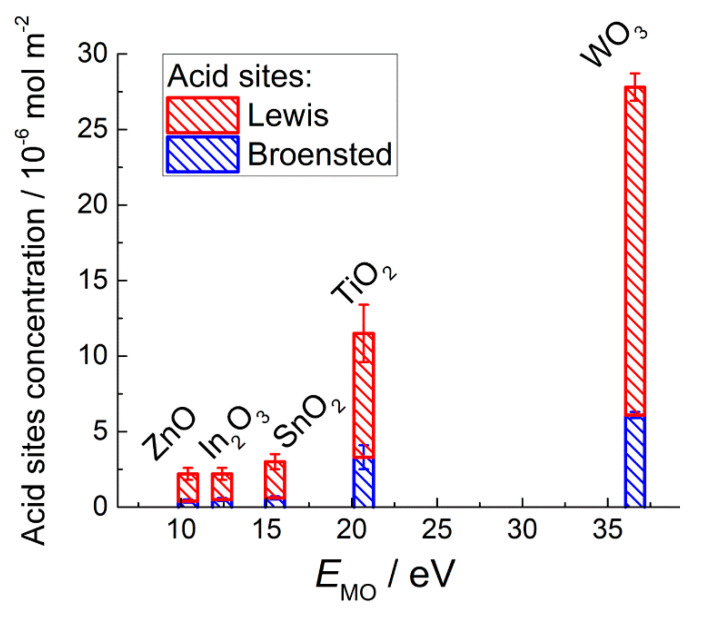
Concentration of Brønsted and Lewis acid sites at the surface of nanocrystalline n-type MOS synthesized at 300 °C (ZnO, In_2_O_3_, SnO_2_, WO_3_) and 700 °C (TiO_2_) in relation to metal-oxygen bond energy. Adapted with permission from reference [78]. Copyright 2021 Elsevier.

**Figure 13 sensors-21-02554-f013:**
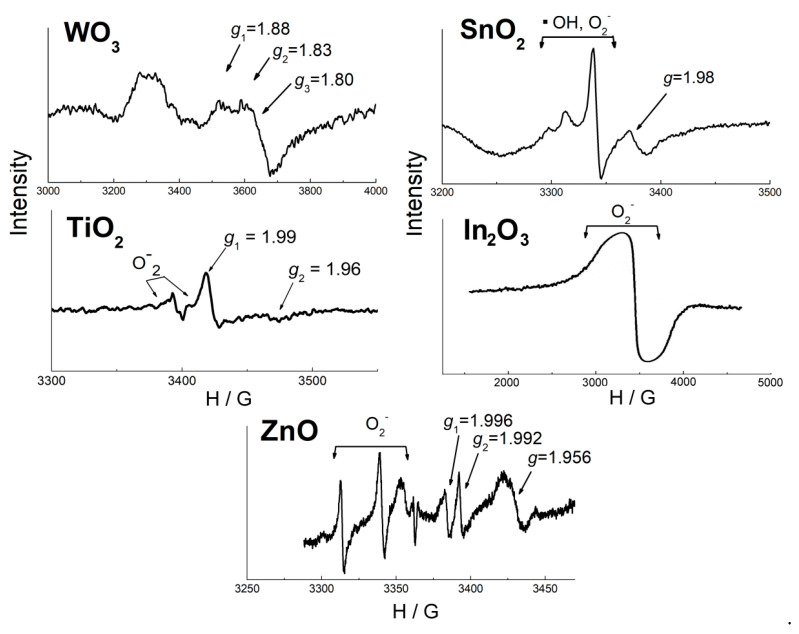
EPR spectra of nanocrystalline *n*-type MOS. Adapted with permissions from ref. [79] copyright 2014 American Chemical Society; ref. [95] copyright 2013 American Chemical Society; ref. [96] copyright 2021 Springer Nature; ref. [101] copyright by the authors; ref. [102] copyright 2013 Elsevier.

**Figure 14 sensors-21-02554-f014:**
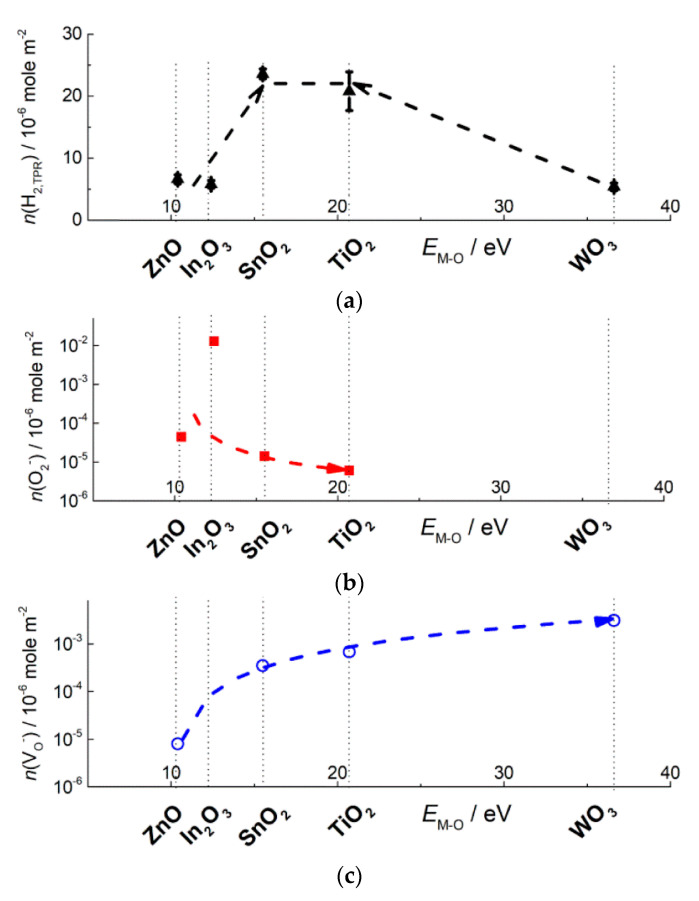
Concentration of active sites at the surface of nanocrystalline *n*-type MOS in relation to metal-oxygen bond energy: oxidizing sites estimated from H_2_ consumption in TPR at temperature below 300 °C (**a**), ionosorbed oxygen O_2_^−^ determined by EPR (**b**), and donor sites (V_O_^−^) determined by EPR (**c**). The values are taken from Table 2.

**Figure 15 sensors-21-02554-f015:**
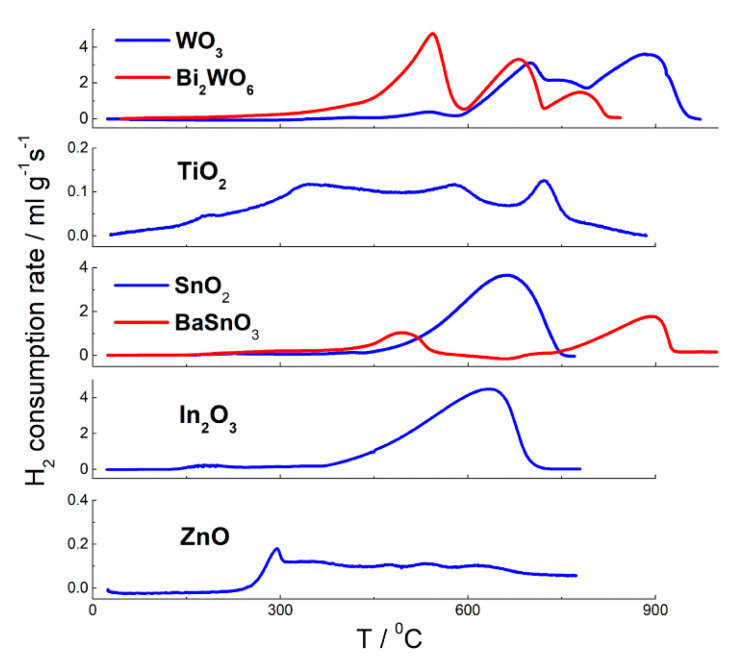
Temperature plots of hydrogen consumption rate during TPR of nanocrystalline *n*-type MOS. Adapted with permissions from ref. [63] copyright 2021 Elsevier; ref. [75] copyright 2018 Elsevier; ref. [76] copyright by the authors (CC BY); ref. [78] copyright 2021 Elsevier; ref. [79] copyright 2014 American Chemical Society.

**Figure 16 sensors-21-02554-f016:**
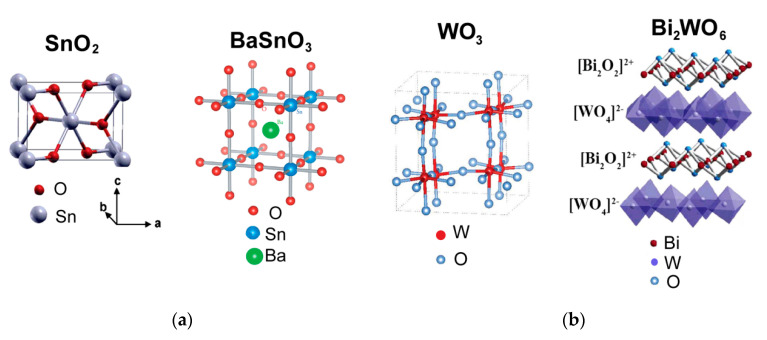
Unit cells of rutile-like tetragonal SnO_2_, perovskite-like cubic BaSnO_3_ (**a**), monoclinic WO_3_ and Aurivillius structure of Bi_2_WO_6_ (**b**). Adapted with permissions from references [61,63,113]. Copyrights 2013 American Physical Society, 2021 Elsevier, 2014 Elsevier.

**Figure 21 sensors-21-02554-f021:**
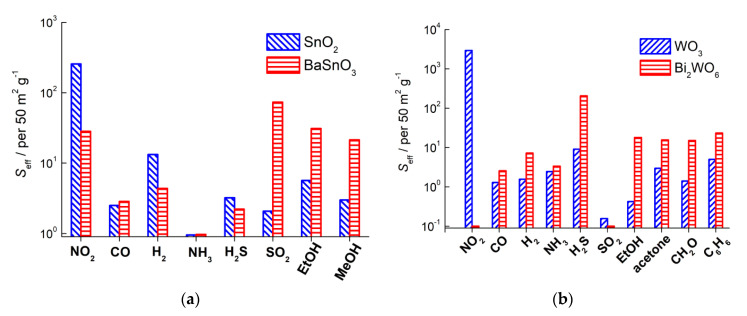
Sensitivity of nanocrystalline SnO_2_ and BaSnO_3_ to NO_2_ (2 ppm) at 100 °C, CO (50 ppm), NH_3_ (20 ppm), H_2_S (2 ppm), H_2_ (100 ppm), SO_2_ (10 ppm), ethanol (20 ppm), methanol (20 ppm) at 300 °C (**a**) [77]; and sensitivity of nanocrystalline WO_3_ and Bi_2_WO_6_ to NO_2_ (1 ppm) at 100 °C, ethanol (20 ppm) at 150 °C, SO_2_ (2 ppm), formaldehyde (400 ppb), H_2_S (2 ppm) at 250 °C, CO (20 ppm), H_2_ (50 ppm), NH_3_ (20 ppm), acetone (2 ppm), benzene (2 ppm) at 300 °C (**b**) [63].

**Figure 22 sensors-21-02554-f022:**
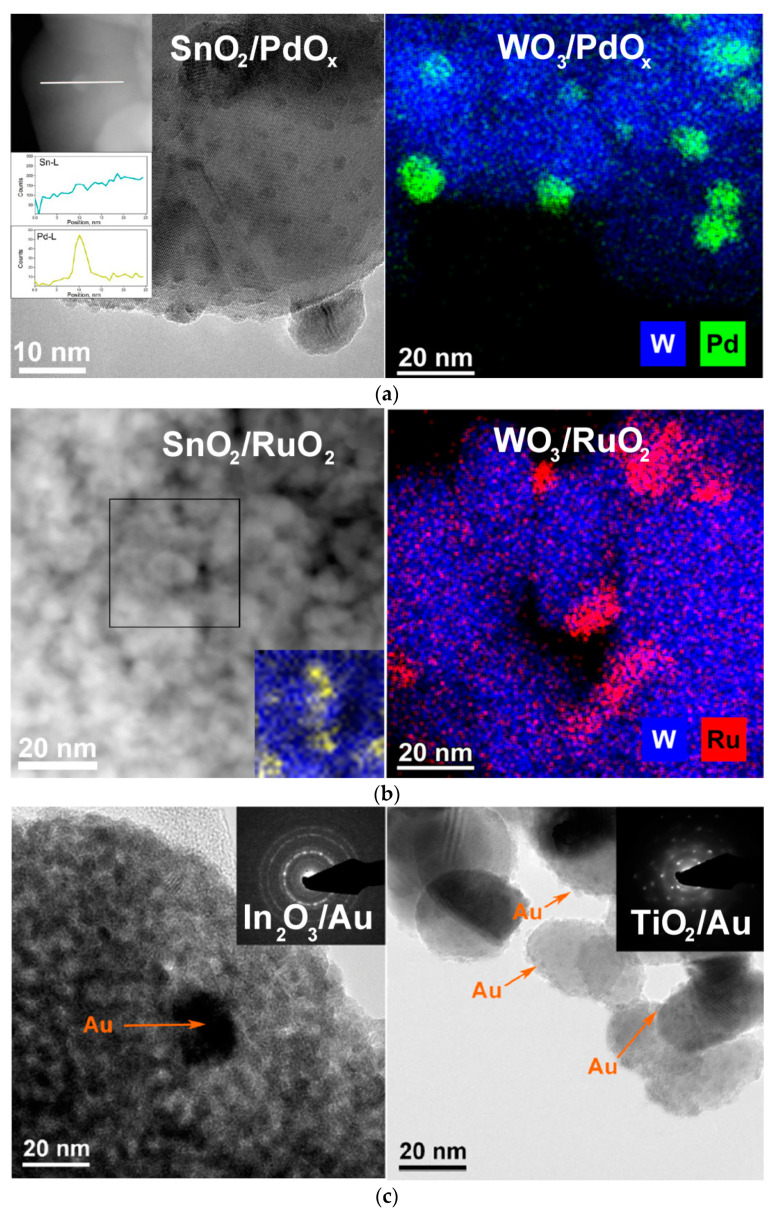
TEM micrographs, STEM images and EDX elemental maps of nanocrystalline SnO_2_ and WO_3_ functionalized by PdO_x_ (**a**) and RuO_2_ (**b**); and TEM micrographs with electron diffraction patterns of In_2_O_3_ and TiO_2_ functionalized by Au (**c**). Adapted with permissions from ref. [75] copyright 2018 Elsevier; ref. [78] copyright 2021 Elsevier; ref. [95] copyright 2013 American Chemical Society.

**Figure 23 sensors-21-02554-f023:**
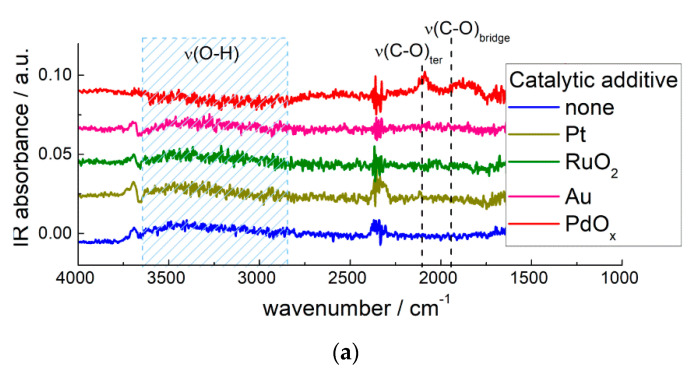
DRIFT spectra of pristine and functionalized by noble metal additives nanocrystalline SnO_2_ (**a**) and WO_3_ (**b**) exposed to CO (100 ppm (**a**); 200 ppm (**b**)) at room temperature for 1 h. Adapted with permissions from ref. [75] copyright 2018 Elsevier; ref. [118] copyright 2015 American Chemical Society.

**Figure 24 sensors-21-02554-f024:**
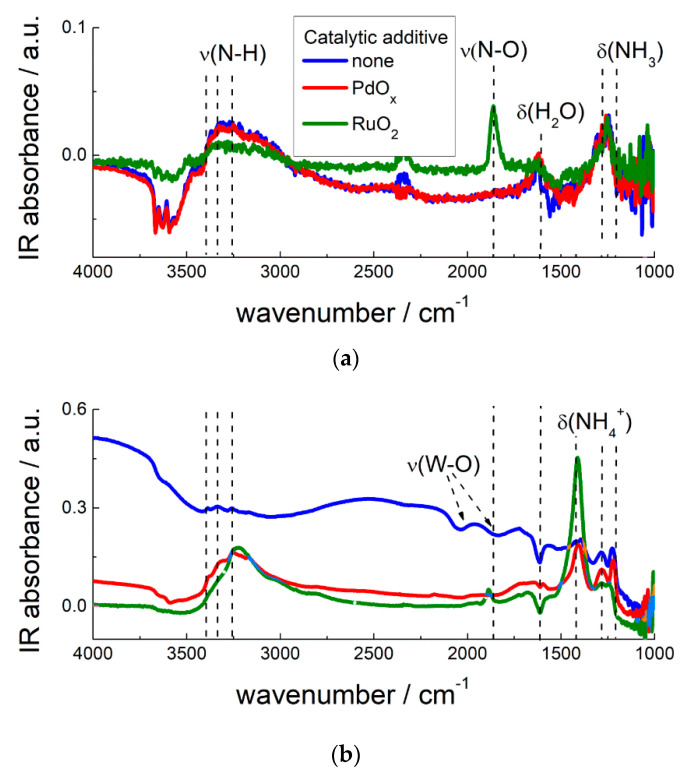
DRIFT spectra of pristine and functionalized by noble metal additives nanocrystalline SnO_2_ (**a**) and WO_3_ (**b**) exposed to NH_3_ (100 ppm (**a**); 200 ppm (**b**)) at 200 °C for 1 h. Adapted with permissions from ref. [75] copyright 2018 Elsevier; ref. [118] copyright 2015 American Chemical Society.

**Figure 25 sensors-21-02554-f025:**
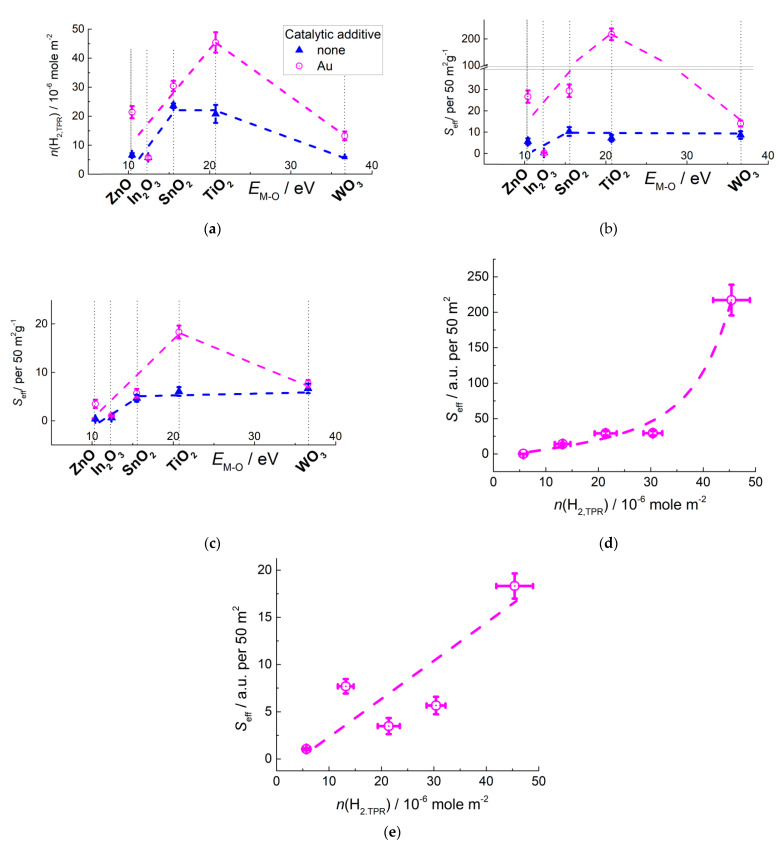
Concentration of oxidizing sites at the surface of pristine and Au-functionalized *n*-type MOS estimated from H_2_ consumption in TPR at temperature below 300 °C (**a**) in comparison with the sensitivity to 20 ppm of acetone (**b**) and 20 ppm of methanol (**c**) as a function of metal-oxygen bond energy in MOS. Sensitivity of Au-functionalized MOS to 20 ppm of acetone (**d**) and 20 ppm of methanol (**e**) in relation to of oxidizing sites concentration (**d**,**e**). Operation temperature of sensors was 250–300 °C for pristine MOS and 150–225 °C for Au-functionalized MOS. Adapted with permission from ref. [78]. Copyright 2021 Elsevier.

**Table 1 sensors-21-02554-t001:** Parameters of *n*-type metal oxide semiconductors (MOS): cationic radius r, electronegativity χ and ionic-covalent parameter ICP; metal-oxygen bond energy *E*_M-O_ and bond distance *d*_M-O_; negative formation enthalpy per oxygen atom −1/*x*·Δ_f_H; bandgap width *E*_g_.

MOS	r, Å [59]	χ, ^1^ P.u. [54]	ICP ^2^ [54]	*E*_M-O_^3^, eV [55]	*d*_M-O_, Å	−1/*x*·Δ_f_H [55], eV	*E*_g_ [54], eV
ZnO	0.74	1.65	0.45	10.4	1.96–1.98 [57]	3.6	3.4
In_2_O_3_	0.79	1.78	0.53	12.4	2.18 [60]	3.2	2.8
SnO_2_	0.69	1.96	0.61	15.5	2.02 [61]	6.0	3.6
TiO_2_	0.61	2.01	0.52	20.7	1.95 (rutile) 1.96 (anatase) [62]	9.8	3.4
WO_3_	0.58	2.36	0.49	36.6	1.74–2.17 [63]	8.7	2.8

^1^ P.u. is Pauling units; ^2^ Ionic-covalent parameter (dimensionless) correlates to Pearson’s hardness of cations [54]; ^3^ Bond energy in metal oxides MO_x_ estimated from thermodynamic parameters [55]: *E*_M-O_ = {−**Δ**_f_H°(MO_x_) + Δ_sub_H°(M) + *x*/2·Δ_dis_H°(O_2_) + *x*·Δ_el.af._H°(O) + ΣΔ_ion._H°(M)}/(*x*·CN_O_), where Δ_sub_H°(M)—sublimation enthalpy of metal, Δ_dis_H°(O_2_)—dissociation enthalpy of oxygen molecule, Δ_el.af._H°(O)—electron affinity of oxygen atom, ΣΔ_ion._H°(M)—sum of ionization potentials of metal cation, CN_O_—coordination number of oxygen in the oxide.

**Table 2 sensors-21-02554-t002:** Concentration of active sites at the surface of *n*-type MOS obtained at different annealing temperature (*T*_anneal_) and having different mean crystallite size (*d*_XRD_) and specific surface area (SSA) [63,76,77,78,79,94,96].

MOS	*T*_anneal_, °C	*d*_XRD_, nm	SSA, m^2^/g	Active Sites Concentration, 10^−6^ mole/m^2^				
				Acid Sites ^1^		Oxidizing Sites		Donor Sites V_O_^− 3^
				Weak (Broensted)	Strong (Lewis)	Total *n* (H_2,TPR_) ^2^	O_2_^− 3^	
ZnO	300	11–13	45	0.4 ± 0.1	1.8 ± 0.4	6.6 ± 0.5	4.4 × 10^−5^	8 × 10^−6^
In_2_O_3_	300	7–8	100	n.d. ^4^	n.d. ^4^	5.8 ± 0.4	1.3 × 10^−2^	-
	500	16–19	35	0.5 ± 0.1	1.7 ± 0.4	4.1 ± 0.8	1.5 × 10^−2^	
SnO_2_	300	3–5	95	0.6 ± 0.1	2.4 ± 0.5	23.6 ± 0.8	1.4 × 10^−5^	3.5 × 10^−4^
	500	10–12	25	0.5 ± 0.2	1.5 ± 0.4	14.0 ± 1.2		
	700	16–20	10	0.5 ± 0.2	1.2 ± 0.5	12.6 ± 2.6		
BaSnO_3_	500	20–22	8	0.4 ± 0.2	0.7 ± 0.3	12.2 ± 2.8	2.5 × 10^−4^	-
TiO_2_	700	27–30 (rutile), 38–46 (anatase)	5	3.3 ± 0.8	8.2 ± 1.9	20.8 ± 3.1	6 × 10^−6^	6.8 × 10^−4^
WO_3_	300	7–9	35	6.1 ± 0.2	21.7 ± 0.9	5.4 ± 0.6	-	3.1 × 10^−3^
	450	19–22	9	5.1 ± 0.9	15.6 ± 1.2	-		
	600	23–25	4	3.1 ± 0.6	8.9 ± 2.0	-		
Bi_2_WO_6_	300	14–15	9	0.8 ± 0.3	1.7 ± 0.6	16.4 ± 2.0	-	-

^1^ Evaluated by TPD of ammonia. ^2^ Evaluated by TPR with hydrogen. ^3^ Evaluated by EPR. ^4^ No data for TPD from In_2_O_3_ annealed at 300 °C because of ammonia oxidation caused by residual nitrate species [78,97].

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
