# Peer review of "The Key Role of Active Sites in the Development of Selective Metal Oxide Sensor Materials"

_sensors, 2021, doi:10.3390/s21072554_

Round 1

Reviewer 1 Report

In "the key role of active sites in the development of selective metal oxide sensor materials" authors report a very accurate and well described review about the employment of nanocrystalline n-type metal oxides (In2O3, ZnO, SnO2, TiO2, WO3) as chemical sensors focusing on the materials composition, concentrations of active sites at the surface, and gas sensing behavior. In the reviewer opinion the manuscript is well written and detailed, there is only a minor consideration that should be taken into consideration to authors, to please check paragraphs numbers, since it changes from paragraph 3.2 to 5.3. 

- Considering the introduction of the manuscript, in the reviewer opinion it would be useful if authors may include a figure or scheme depicting the different sensing mechanisms toward reducing and oxidizing molecules and oxygen. Furthermore it would be helpful if introduction could be separated into different paragraphs to improve the readability.
- Please define cus acronym in the Figure 2  (reviewer thinks it is referred to coordinatively unsaturated sites ).
- line 293-296 - The binding energies in the text seem to do not match the dotted lines in the corresponding figure (Figure 5, curves are centered at 530.5, 531.75) please check them.
- line 343: Appendix A ( check typo).
- It would be better if the lettering of Figures with multiple panels is the same for all the figures. E.g. lettering (a) and (b) should be displaced below the corresponding panel. Please check the lettering of Figure 11, 13, 16, 19, 21, 22
- Considering Figure 18, in the reviewer opinion responses should be normalized in order to be compared since measurements were performed at different gas concentrations. May the authors normalize the data by dividing the response by the relative gas concentration (ppm). Please note that this normalization would be indeed strictly correct only if the sensors are in their linear range, although it can improve the readability of the figure ( considering the caption too).
- In figure 20 and 21 panel a and b should be cited as (a) and (b); e.g. (100 ppm, (a); 200 ppm, (b)) 

Author Response

Reviewer comment 1: please check paragraphs numbers, since it changes from paragraph 3.2 to 5.3.

Author response 1: We thank the Reviewer for notice. The numbering of paragraphs was corrected in the revised manuscript.

Reviewer comment 2: Considering the introduction of the manuscript, in the reviewer opinion it would be useful if authors may include a figure or scheme depicting the different sensing mechanisms toward reducing and oxidizing molecules and oxygen. Furthermore it would be helpful if introduction could be separated into different paragraphs to improve the readability.

Author response 2: The figures 1-3 were added into the revised manuscript depicting the interaction with oxygen, reducing and oxidizing gases within the chemisorption model and oxygen vacancy model. The Introduction was subdivided into subsections in the revized manuscript.

Reviewer comment 3:  Please define cus acronym in the Figure 2  (reviewer thinks it is referred to coordinatively unsaturated sites ).

Author response 3: The figure caption was revized and the acronym specified.

Reviewer comment 4: line 293-296 - The binding energies in the text seem to do not match the dotted lines in the corresponding figure (Figure 5, curves are centered at 530.5, 531.75) please check them.

Author response 4:  The binding energies in the text were revised in agreement to the figure. In common, the O 1s peak has usually been deconvoluted into two parts, with the main component at lower binding energy attributed to lattice anions, and the smaller one at higher binding energy - to surface oxygen species. The values of binding energies are not strict and may vary depending on the oxide studied and deconvolution precision.

Reviewer comment 5: line 343: Appendix A ( check typo).

Author response 5:  The mistake was corrected.

Reviewer comment 6: It would be better if the lettering of Figures with multiple panels is the same for all the figures. E.g. lettering (a) and (b) should be displaced below the corresponding panel. Please check the lettering of Figure 11, 13, 16, 19, 21, 22

Author response 6:  The lettering of mentioned figures was remade uniformly throughout the manuscript.

Reviewer comment 7: Considering Figure 18, in the reviewer opinion responses should be normalized in order to be compared since measurements were performed at different gas concentrations. May the authors normalize the data by dividing the response by the relative gas concentration (ppm). Please note that this normalization would be indeed strictly correct only if the sensors are in their linear range, although it can improve the readability of the figure ( considering the caption too).

Author response 7:  We believe that normalization of the sensitivity to same gas concentrations would be improper. The power dependence of sensor response on target gas concentration is usually the case S ~ Ca, and the power a may vary significantly from unity. In the reviewed works, these dependences were not studied for all the tested gases. Moreover, it seems unnecesarry, because the threshold limit concentration varies significantly for different analytes. In our works, we aimed at comparing the sensitivities to the such concentrations of analytes that are close to threshold limit values at a working place. This is the origin of different concentrations mentioned in the figure caption. 

Reviewer comment 8: In figure 20 and 21 panel a and b should be cited as (a) and (b); e.g. (100 ppm, (a); 200 ppm, (b))

Author response 8: The citation of figures panels was corrected.

Reviewer 2 Report

The paper aims at providing an overview on metal oxide semiconductors (MOS) for sele tive gas sensors. The authors propose a summary of their recent works on the determination revelation of active sites at the surface of gas sensitive MOS-based materials and the impact of surface sites to the adsorption and redox interaction upon gas interaction.

In my opinion the review is quite original providing great advances in current knowledge, especially for researcher last approaching to the topic. Quality of the presentation is good; there could be an the overall benefit to publishing the work, readers of the journal could be highly stimulated.

Just as a suggestion: the authors could add some conclusion remarks highlighting future perspectives and challenges that scientific research in this field is facing up.

After this the paper can be accepted for publication.

Author Response

Reviewer comment: Just as a suggestion: the authors could add some conclusion remarks highlighting future perspectives and challenges that scientific research in this field is facing up.

Authors response: We are gratefull for such an appreciation of the manuscript. Some words on the future research and challenges was added at the end of Conclusions section in the revised manuscript.